# AutoNumerics-Zero:
# Automated Discovery of State-of-the-Art Mathematical Functions

Esteban Real [1]   Mirko Rossini [2]   Connal de Souza [2]   Manav Garg [2]   Moritz Firsching [1 3]
Quoc V. Le [1]   Yao Chen [4]   Akhil Verghese [4]   Ekin Dogus Cubuk [4]   David H. Park [2]

## Abstract

Transcendental functions, such as the exponential, are central to scientific computing, yet they cannot be natively calculated by digital hardware. Instead, computers must approximate these functions by combining basic operations, such as $\{+, -, \times, \div\}$, using methods like Taylor series. These methods were developed over centuries by mathematicians, who focused on approaches that could attain arbitrary accuracy. However, computers can handle most applications by using only finite-precision types, like *float32*, where any accuracy beyond the type's precision is effectively discarded. We explore, therefore, whether forgoing arbitrary accuracy can lead to the discovery of more efficient approximations. The evolutionary method of symbolic regression is particularly suitable, as it can search for arbitrary operation combinations and can optimize non-differentiable objectives, such as the number of operations used. Our results show that evolution can discover computer programs that outperform established methods in this setting, despite having no prior mathematical knowledge beyond the calculation of the basic operations. Starting from empty code, symbolic regression constructs programs representing novel mathematical expressions. In particular, we discovered a 10-operation program that approximates the exponential function to 14 significant figures, exceeding the accuracy of previously known approximations of this size by more than 6 orders of magnitude.

[1]Google DeepMind [2]Google [3]Google, Paradigms of Intelligence Team [4]For work done while at Google. Correspondence to: Esteban Real <ereal@google.com>.

*Proceedings of the 43$^{rd}$ International Conference on Machine Learning*, Seoul, South Korea. PMLR 306, 2026. Copyright 2026 by the author(s).

## 1. Introduction

The numerical calculation of transcendental functions—such as exponentials, logarithms, and trigonometric functions—is fundamental to the applied sciences. These functions are pervasive in computationally intensive simulations (Todorov et al., 2012; Voter, 2007; Stegailov et al., 2019). To evaluate them, computers rely on a small set of hardware-executable operations, such as addition and multiplication. Since there exists no finite sequence of such operations that can yield exact results, transcendental functions must be approximated. This is done with various methods, which can attain arbitrary accuracy with enough operations. A simple example is the truncated Taylor series, which improves in accuracy as more terms are included. However, modern hardware typically utilizes finite-precision data types, such as *float32* (Zuras et al., 2008), which have proven sufficient for most applications. In this context, accuracy beyond a data type's precision is effectively wasted. Motivated by this observation, this paper investigates whether optimizing for high—but finite—accuracy targets can lead to more efficient approximations. Our core assumption is that if a discovered approximation is efficient and sufficiently accurate for most applications, the discovery cost will be amortized by the efficiency gains in all future uses.

We hypothesize that large-scale symbolic regression, over the space of computer programs, is a promising approach to the discovery of finite-accuracy function approximations. Symbolic regression is an evolutionary search process that looks for expressions (in our case, computer programs) that optimize given quality metrics (in our case, the program's accuracy and length) (Koza, 1992; Schmidt & Lipson, 2009; La Cava et al., 2021). Evolutionary search is particularly suitable for our purpose because it can discover arbitrary combinations of the operations $\{+, -, \times, \div\}$, unlike traditional approaches, which must conform to a fixed form (*e.g.*, a polynomial form in the case of the Taylor series). Representing the functions as *programs* (*e.g.*, Figure 1, right)—instead of formulas—has the advantage that they can specify the reuse of intermediate values, potentially leading to shorter calculations. The evolutionary process improves a population of programs by iteratively keeping only the best

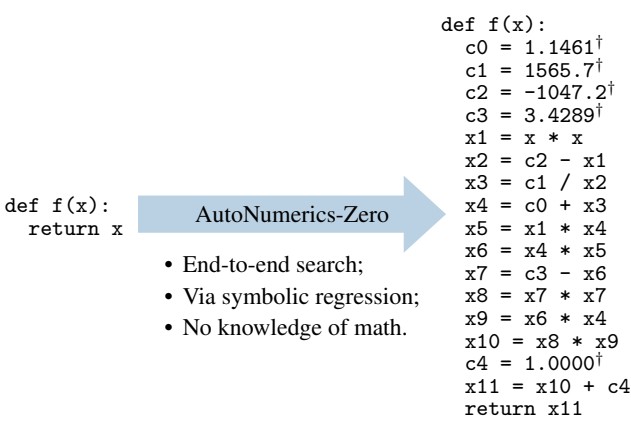

```
def f(x):
  return x
```

AutoNumerics-Zero

- End-to-end search;
- Via symbolic regression;
- No knowledge of math.

```
def f(x):
  c0 = 1.1461†
  c1 = 1565.7†
  c2 = -1047.2†
  c3 = 3.4289†
  x1 = x * x
  x2 = c2 - x1
  x3 = c1 / x2
  x4 = c0 + x3
  x5 = x1 * x4
  x6 = x4 * x5
  x7 = c3 - x6
  x8 = x7 * x7
  x9 = x6 * x4
  x10 = x8 * x9
  c4 = 1.0000†
  x11 = x10 + c4
  return x11
```

*Figure 1.* AutoNumerics-Zero searches for programs from scratch. Left: Empty program used as the starting point of the search. Right: Example of a final program discovered. †Rounded.

and generating more by modifying them with small random edits called *mutations*. In this work, we ensure that these mutations are simple and random enough so that they cannot introduce any knowledge of mathematics into the search process (*e.g.*, they can introduce a new line of code that adds two variables, but which two variables are selected must be a random choice). This, together with the fact that we start each evolutionary experiment with *empty* programs (Figure 1, left), results in a "zero-knowledge" search process[1]: it possesses no prior mathematical knowledge of asymptotics, perturbation theory, or any other numerical approximation technique; in particular, it does not use expansion rules, compute derivatives, or pre-train on a corpus of human-designed functions. This renders the process free from the bias of existing approaches, potentially leading to novel discoveries. Thus, it provides a complementary approach to methods heavily reliant on training data, such as large language models. More broadly, this paper does not seek to introduce a new search algorithm. Instead, it combines various known stochastic-optimization techniques into a method we dub *AutoNumerics-Zero*. This way, we rigorously demonstrate a *practical* systems application of symbolic regression to transcendental function approximation—for the first time.

We verify our hypothesis by showing that programs discovered with AutoNumerics-Zero compare favorably to those found with multiple baseline methods, through several examples of target transcendental functions. These target functions were chosen to illustrate different challenges to function approximation, which will be detailed in Section 4. We do not claim, however, that this evolutionary method can surpass baselines for every target function. In fact, traditional methods do not work well in every case either; indeed,

entire chapters of introductory textbooks are dedicated to the variety of techniques suitable in different situations (Bender & Orszag, 1999). For every target function, we compare the best discovered approximations against baselines that are expected to do well for that function. We draw such baselines from established methods that have been demonstrated to produce high accuracy numerical approximations, including Chebyshev polynomials, Padé approximants, and minimax optimization (Mason & Handscomb, 2002; Baker Jr. & Gammel, 1961; Wolfram Research, Inc., 2023). In cases where our method shows the most salient advantages, we verify our claims by mathematically proving tight error bounds; the proofs are fairly automated and can be applied more generally.

With the stated methodology, we find that searching for finite accuracy targets does indeed result in improved approximations. Symbolic regression discovers novel, practical, and highly accurate programs, which can significantly surpass the baselines. In particular, we consider our discovered exponential function programs in Section 4.1 to be noteworthy, as they are measured against especially strong baselines—the exponential is a prominent, well-behaved, and thoroughly studied function; nevertheless, we are able to mathematically demonstrate sizable accuracy improvements for the same number of operations (Figure 4). Further, the evolutionary approach permits multiple extensions of our work that would not be possible with traditional methods. These are mentioned in the Discussion Section and touched upon throughout the Results Section. For example, while most of this paper is hardware-agnostic, Section 4.6 specializes our method to optimize for a specific hardware architecture, trading generality for efficiency. This way, we found an approximation more than three times faster than baselines, as evolution engineered programs that trigger unusual but beneficial compilation paths (Section 4.6).

In summary, our contributions are:

- AutoNumerics-Zero, a zero-knowledge symbolic regression method that combines traditional techniques to approximate transcendental functions;
- a rigorous demonstration that, when aiming for high-but-finite accuracy, AutoNumerics-Zero can surpass established baselines;
- discovered programs for exponential calculation that are much more accurate than previously known mathematical expressions for a given number of operations; and
- evidence of extensibility of the method to other functions, operation sets, and the potential to trade generality for efficiency by specializing to a given hardware architecture.

Our open-sourced code can be found online[2].

**Conflict of Interest Disclosure** The authors are employed by Google, which leads the development of JAX/XLA, technologies which are central to brief sections of this paper.

---

[1]On a bounded interval, which is then generalized to the real line through the standard range reduction technique.

[2]https://github.com/google-deepmind/autonumerics_zero

## 2. Related Work

### 2.1. Discovering New Mathematics

Recent work has discovered state-of-the-art mathematical algorithms for matrix multiplication (Fawzi et al., 2022) and sorting (Mankowitz et al., 2023). This was done by searching at scale using reinforcement learning (RL) over small-input problems (*e.g.*, multiplying 4x4 matrices) and then generalizing to larger inputs through analytical means. To the best of our knowledge, our work is the first to discover state-of-the-art mathematical algorithms through evolutionary search, with a critical difference from the aforementioned RL findings: rather than searching at a small input size and generalizing analytically, we search directly at practical input sizes (*e.g.*, 32-bits for floating-point values). This allows the search method to explore the large-scale structure of the solutions too. We opted for evolutionary computation primarily for its simplicity; re-formulating this task within a reinforcement learning framework remains an open direction.

### 2.2. Relationship to Symbolic Regression Work

Our work is an example of *symbolic regression*, a method that automatically discovers a symbolic relationship that fits given data. Symbolic regression has been used to find physics formulas by means of evolutionary computation (Schmidt & Lipson, 2009; Long et al., 2018; Sahoo et al., 2018; Wang et al., 2019) or non-evolutionary methods (Brunton et al., 2016; Rudy et al., 2017; Schaeffer, 2017; Udrescu & Tegmark, 2020; Biggio et al., 2021). These studies demonstrate that their respective approaches can parse simulated data to recover well-known equations. Despite its success on complex systems, symbolic regression has not been used to discover *previously unknown* physical laws, to the best of our knowledge. Using symbolic regression on true experimental data has only been done in very specialized situations (Stanislawska et al., 2012; La Cava et al., 2016), where validating the correctness of the discovered models is difficult.

In contrast, our work discovers previously unknown symbolic relationships that we verify with mathematical proofs. This is made possible because the task is defined *in silico* (*e.g.*, how a computer should calculate an exponential), as opposed to "in nature" (*e.g.*, how gravitation works). Thus, in our case, the data used during the search process is free from experimental error.

Previous studies have attempted the *in silico* task of discovering arbitrary formulas from data, but the solutions were known ahead of time and the data was generated directly from them (Uy et al., 2011; Kusner et al., 2017; Petersen et al., 2021). Because their goal was to develop the search methodology, these studies focused on simple formulas. Fu-

ture work could explore whether these more sophisticated methods can be applied to our task.

### 2.3. Relationship to Program Discovery Methods

Our work uses genetic programming (GP) (Koza, 1992; Banzhaf et al., 1998; Spector et al., 1999), and more generally, evolutionary computation (Fogel et al., 1966; Holland, 1992), which has found success in hardware design (Lohn et al., 2005), software engineering (Le Goues et al., 2011), and medicine (Cortacero et al., 2023), among others. The field provides multiple algorithms but relatively little guidance as to their applicability (Orzechowski et al., 2018). Thus, we chose to use a simple method, staying as close as possible to existing techniques.

A related field is *superoptimization* (Joshi et al., 2002; Bansal & Aiken, 2006; Schkufza et al., 2013), which improves a program by searching through code transformations. In particular, Schkufza et al. (2014) reduced the compute cost of floating-point programs at the expense of some accuracy. Conversely, Panchekha et al. (2015) improved accuracy at large costs in compute. Relatedly, Damouche & Martel (2017) enhanced the precision of iterative methods, but they did not search for transcendental function expressions nor did they seek to optimize compute cost. In contrast, AutoNumerics-Zero improves both objectives, accuracy and computational efficiency. It can do this because it explores the space of all programs. While superoptimization requires a correct program as the starting point of the search, our approach finds novel programs starting from empty code. Future work may explore to what extent benefits from both approaches may compound.

## 3. Methods Outline

We evolve programs to approximate a target function (*e.g.*, $f(x) = 2^x$), aiming for two objectives: few operations and high accuracy, with the accuracy measured as $a = -\log_{10}(E)$, where $E$ is the maximum error over the function's domain. This section outlines the evolutionary search process that discovers the programs. After this search is complete, the discovered programs are tested thoroughly on unseen examples (Appendix A.10). Moreover, for the best discovered programs, we mathematically prove tight error bounds using interval arithmetic (Appendix B).

**Evolutionary Search.** To discover approximations of target functions, we follow a symbolic regression paradigm (Schmidt & Lipson, 2009; Long et al., 2018), to gradually improve a population of candidate programs. The programs are written as imperative code where each instruction either defines a coefficient or applies an operation from $\{+, -, \times, \div\}$; Figure 1 in the Introduction showed an example. The population starts out with only empty programs

(Figure 1, left) and improves them in cycles through a nested optimization process, ending in accurate and compact approximations (*e.g.*, Figure 1, right). One cycle of the process is shown in Figure 2 below. A large-scale outer loop uses genetic programming techniques (Koza, 1992; Banzhaf et al., 1998) to discover the symbolic structure of the program (*e.g.*, akin to "$\texttt{return } x \times c_0 + c_1$"), while an inner loop optimizes the floating-point coefficients (*e.g.*, "$c_0$" and "$c_1$"). Symbolic regression is a loose term for this general approach; we will now outline our choices for the algorithms used in each of the two loops, the program representation, and the rules ("mutations") employed to generate new programs. Section 4.8 presents ablations for our method's components.

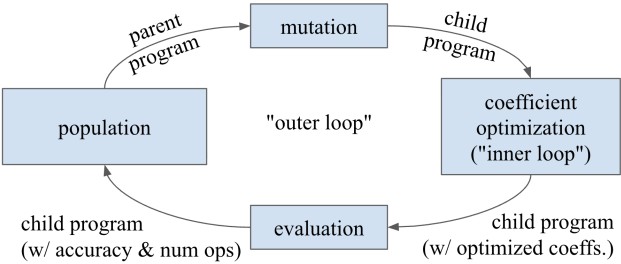

*Figure 2.* Evolutionary search. This 4-stage cycle is iterated to gradually improve a population of programs over time.

We execute multiple instances of the cycle asynchronously and in parallel. Generally, we run 100–10k processes for 1–4 days on a custom cluster of Skylake cores, which we estimate could be reproduced on 100 modern high-end CPUs. Further detail can be found in Appendix A.9.

---

**Method 1** Population Selection Stage: dNSGA-II Outline

---

**input** a worker pool $\mathbb{W}$ of size $\mathtt{W}$
**input** a sample size parameter $\mathtt{S} \ll \mathtt{W}$
**output** a set of $\mathtt{W}$ evolved programs.
   # We use Python notation for operations on lists:
   # "+" = concatenate; "*" = repeat.
  **parallel-for** $\mathtt{w} \in \mathbb{W}$ **do async**
    **while** $\mathtt{w}.\mathtt{is\_running}$ **do**
      **if** first iteration **then**
        # For the first generation, start from scratch.
        $\mathcal{S} = [\mathbb{1}] * 2\,\mathtt{S}$  # List of identity programs.
      **else**
        $\mathcal{S} = $ **receive** $2\,\mathtt{S}$ programs from random workers in $\mathbb{W}$
      **end if**
      $\mathcal{P} = \texttt{SelectNearPareto}(\mathcal{S})$  # Selects $\mathtt{S}$ parents from $\mathcal{S}$.
      $\mathcal{C} = [\,]$  # Generated children.
      **for** $p$ **in** $\mathcal{P} + \mathcal{P}$ **do**  # Cycle through parents twice.
        $\mathcal{C} = \mathcal{C} + [\texttt{MutateAndEvaluate}(p)]$
      **end for**
      **assert** $|\mathcal{C}| = 2\,\mathtt{S}$
      **send** $\mathcal{C}$ to other workers in $\mathbb{W}$
    **end while**
  **end parallel-for**
  **return** latest programs evaluated

---

**Population selection stage** (Figure 2, starting at the left-most box): As indicated above, the population of programs is improved in cycles. Each cycle begins by pruning the population, selecting good *parent* programs to keep and discarding the rest. For this, we use our custom variant of the popular NSGA-II algorithm. We chose NSGA-II because it is known to handle multiple objectives well (Deb et al., 2002), in our case by favoring programs near the high-accuracy–few-operations Pareto front. In NSGA-II, a "$\texttt{SelectNearPareto}$" operation acts on the entire population, choosing programs close to the front while ensuring even separation of programs along the front. Our custom variant, dubbed dNSGA-II, is convenient for a distributed cluster, as it avoids the need for a centralized location for the population (Method 1) and permits updating the population asynchronously. (Details in Appendix A.3.)

**Mutation stage** (Figure 2, top-most box): Each selected parent is then randomly modified with small edits called *mutations* to generate *child* programs. The mutations operate on the *compute graph* representation of a program (Koza, 1992). The compute graph is a directed graph in which (a) input vertices represent the program's input and coefficients, (b) intermediate vertices represent the operations, and (c) a single output node represents the program's output; Figure 3 (top) is an example. A mutation takes one of the following three actions, chosen at random: (1) introduction of a new vertex with an operation randomly sampled from $\{+, -, \times, \div\}$ at a random location in the graph, (2) deletion of a random vertex, or (3) random reconnection of a randomly chosen edge (Figure 3, bottom).

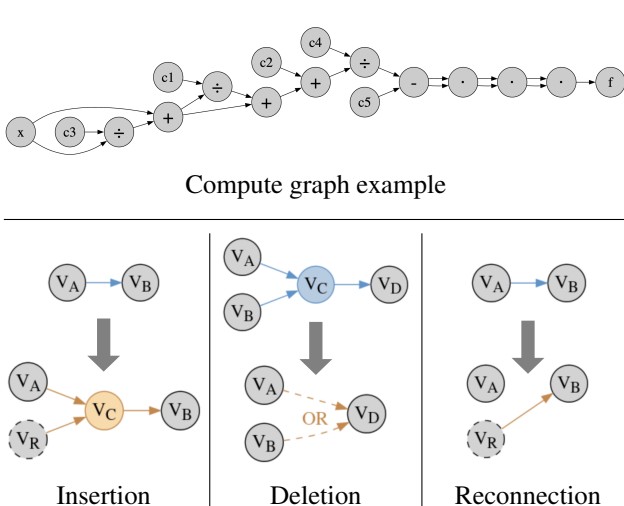

Compute graph example

Insertion | Deletion | Reconnection

*Figure 3.* The mutations act on the compute graph (top), performing one of three actions at random (bottom row). "$V_R$" denotes a random preexisting vertex elsewhere in the graph. "OR" indicates a random choice between edges. When mutating, all choices are random (*e.g.*, an insertion introduces a random operation from $\{+, -, \times, \div\}$ at a random position in the graph).

**Coefficient optimization stage** (Figure 2, right-most box): Once mutated, each child has its coefficients optimized for high accuracy in a CMA-ES inner loop. CMA-ES is a continuous optimization algorithm, like gradient descent, except that it is gradient-free and less sensitive to local optima; it tends to work well with low parameter counts (Hansen & Ostermeier, 2001; Hansen et al., 2015). The optimization loss is computed over a batch of correct input–output examples. *E.g.*, if the goal is to discover a fast approximation to $f(x) = 2^x$, the labeled examples would be $(x, 2^x)$ pairs that are calculated with another (slower) method, such as a very long Taylor series. These examples are sampled randomly from the target function's domain. (Details in Appendix A.5.)

**Evaluation stage** (Figure 2, bottom-most box): The child program, now with optimized coefficients, is evaluated to obtain its accuracy and the number of operations it uses. The accuracy is estimated by sampling points in the domain of the function and computing the maximum error, as indicated above. The evaluated child is finally added to the population, thus completing one cycle. (Details in Appendix A.6.)

## 4. Results

We will now apply AutoNumerics-Zero to search for high-but-finite accuracy approximations to six target functions, comparing our results against the baselines. As mentioned in Section 1, the target functions were chosen *a priori* to present various challenges to the method and the baselines were selected to represent the best methods currently available. We present the results thoroughly without excluding cases due to search method failure and without tuning the method to each case. The first five cases to follow are hardware-agnostic (much like a Taylor function is hardware-agnostic) while the last case explores a hardware-aware scenario. At the end, we will mathematically verify the discovered programs and present method ablations.

### 4.1. Exponential Function

SETUP (THE FUNCTION'S DOMAIN). We want to approximate $f(x) = 2^x$ over the entire real line $\mathbb{R}$. However, this cannot be done with any finite combination of $\{+, -, \times, \div\}$. This is because any such combination is a rational function and thus grows like $x^k$ for some $k \in \mathbb{Z}$, as $x \to \infty$, while $2^x$ grows much faster, leading to unbounded error. Fortunately, the method of *range reduction* can take any approximation on $(0, 1]$ and extend it to $\mathbb{R}$ (Appendix A.1). Because of this, all the baselines seek an approximation in $(0, 1]$ only. We adopt the same strategy: we apply our zero-knowledge search process to discover the approximation on $(0, 1]$, knowing that the result can be extended to $\mathbb{R}$ afterward.

SETUP (ACCURACY MEASUREMENT). For the accuracy objective, we use $a = -log_{10}(E_{MR})$, where $E_{MR}$ is the maximum relative error. $E_{MR}$ is the generally accepted choice in numerical approximation (Hart, 1978). Combined with range reduction, this metric ensures that $a$ is essentially the number of significant figures guaranteed to be correct across the entire real line.

BASELINES USED. We exhaustively evaluated every established baseline we could identify. A common approach to computing an exponential is through a Taylor expansion ("Poly/Taylor" in Figure 4), written in Horner's form to minimize the number of operations (*i.e.* in the form $c_0 + x(c_1 + x(c_2 + x...))$, see Pan, 1966; Ostrowski, 1954). The method of Padé approximants ("Ratio/Padé" in the figure) derives a rational function from the Taylor coefficients. This rational function has improved convergence properties in the number of operations (Bender & Orszag, 1999). These expansion approaches are suboptimal because they produce exact values at a single point in the domain but generally increasing error with distance from that point. A better approach is to even out the error over the $(0, 1]$ interval. This can be accomplished with a Chebyshev approximation, resulting from writing the function as a linear combination of low-order Chebyshev polynomials (Tchébychev, 1858), and rewriting the result in Horner's form ("Chebyshev" in the figure). A more involved alternative is to minimize the maximum relative error using the Remez algorithm (Remez, 1969) ("Poly/Minimax" and "Ratio/Minimax" in the figure). We also include as baselines continued fractions due to Euler (1748), Gauss (1813), and Macon (1955), the latter having been developed to minimize the number of operations.

THE RESULTS. Figure 4 shows the evolved exponentials

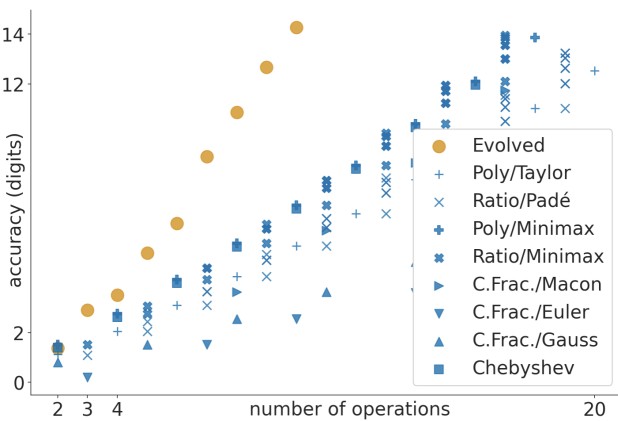

*Figure 4.* Programs evolved from scratch (top row, orange) surpass the baselines (other rows, blue). Each point denotes an approximation of $2^x$. The accuracy (vertical axis) is the number of correct digits guaranteed over the entire real line; so, for example, the approximation corresponding to the highest point in the plot is correct to 14 significant figures for all inputs.

surpass the accuracy of the baselines for the same number of operations. In this plot, the accuracy of the *evolved* programs is an *under*estimate, obtained from error bound proofs (Appendix B.2), while the accuracy of the *baselines* is an *over*estimate, obtained from sampling points on the domain. Therefore, the multi-digit gap between them is mathematically certain. The top evolved program is shown in Figure 5; Appendix Figure 11 shows additional programs.

```
def f(x):
    c1 = 15141.981176922711
    c2 = -250.55247494972059
    c3 = 5783.5330096027765
    x1 = c3 / x
    x2 = x + x1
    x3 = c1 / x2
    x4 = x3 + x2
    x5 = c2 + x4
    c4 = 501.10494991027866
    x6 = c4 / x5
    c5 = -0.99999999999999956
    x7 = x6 - c5
    x8 = x7 * x7
    x9 = x8 * x8
    x10 = x9 * x9
    return x10
```

*Figure 5.* Discovered program for $2^x$ computation with 10 operations and proven maximum relative error under $5.4 \times 10^{-15}$.

### 4.2. Cosine Function

Because of its periodicity, it is sufficient to approximate $\cos(x)$ in $[0, \pi/2]$ and extend to $\mathbb{R}$ by range reduction through shifts and reflections. The measurement of *relative* error is problematic due to the zero of $\cos(x)$, so we circumvent this difficulty by considering the maximum *absolute* error $E_{MA}$ instead; thus the accuracy becomes $a = -\log_{10}(E_{MA})$. With absolute errors, it becomes preferable to distribute the error evenly over the interval, so a strong baseline is the Chebyshev approximation, which was designed to accomplish exactly this. Figure 6 (left) shows that the evolved functions compare favorably.

### 4.3. Airy Function

A challenge to rational approximation is posed by oscillatory functions, like Airy's $\mathrm{Ai}(x)$ for $x < 0$, where evolutionary search still discovers compact and accurate programs. We emphasize the challenge by computing an approximation to $f(x) = \mathrm{Ai}(-kx)$ on $(0, 1]$ while setting $k = 7$ to have roughly two oscillations (the value of $k$ was chosen before running any search experiment). We use the same errors and baselines as in Section 4.2, for the same reasons. Even fairly complex programs can be evolved in this case. For example, the 19-operation evolved function has an estimated accuracy of 4.2, reducing the maximum error by two orders of magnitude compared to the best 20-operation baseline.

### 4.4. Bessel Function

The evolutionary method permits searching for expressions that use arbitrary operations, beyond $\{+, -, \times, \div\}$. The need for other operations often arises when approximating solutions of differential equations near singular points. As an illustration, consider the modified Bessel function $I_{1/2}(x)$ on (0,1], which near $x = 0$ behaves like $\sqrt{x}$. It therefore makes sense to expand our set of operations to $\{+, -, \times, \div, \sqrt{\cdot}\}$, noting that modern computer processors often provide a square-root instruction. As before and for the same reason, we measure accuracy as $a = -\log_{10}(E_{MA})$. Polynomial baselines, like Chebyshev expansions, do not do well because of the behavior near zero. A more appropriate baseline is the method of Frobenius (Bender & Orszag, 1999), which expands about zero guaranteeing the correct asymptotic behavior. Figure 6 (right) shows the comparison between this baseline and our evolved programs.

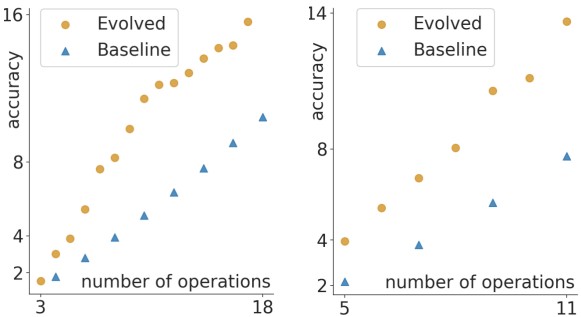

*Figure 6.* Cosine (left) and Bessel function (right). Each plot compares evolved (orange, top row) and baseline (blue, bottom row) approximations. (Evolved programs listed in Appendix D).

______ *A tailored baseline:* *Specifically for $I_{1/2}(x)$, we can hand-design a baseline that narrows the gap to the evolved approximation. First, we factor out the square root to remove the non-polynomial behavior near zero, seeking an approximation to $I_{1/2}(x)/\sqrt{x}$. Observing that the odd Frobenius coefficients vanish, we reparameterize the expression to $\tilde{g}(x) = I_{1/2}(\sqrt{x})/\sqrt[4]{k}$. $\tilde{g}$ can be efficiently approximated with the Chebyshev method after an affine transformation to $(-1, 1]$, the interval where Chebyshev polynomials are orthogonal. This approach, for example, produces a 9-operation approximation with 7.8 accuracy, slightly less accurate and with one more operation than the 8.1-accuracy evolved program. We highlight, moreover, that this manual method required thinking carefully about this specific function, whereas the evolutionary approach is much more automated.*______

### 4.5. Error Function

We attempt to approximate the real-valued $\mathrm{erf}$ on $(0, 2]$ because of the notoriously slow convergence of the Taylor expansion in this domain (Sloane et al., 2003). While our method finds relatively more concise functions at the lowest accuracies, at higher accuracies Padé approximants are better, partly due to the fortuitous vanishing of many terms in the quotient polynomials.

## 4.6. Hardware-aware Exponential Function

SETUP. In this section, we look again for exponential function approximations but in a hardware-aware setup. While so far we have considered hardware-agnostic results, we now trade generality for efficiency by optimizing for a specific hardware architecture. This approach may prove useful, for example, before running a very long simulation on a given supercomputer with known hardware. In this case, some of the simulation's compute budget may be borrowed to first improve the efficiency of frequently used functions, thus reducing the overall total cost. Our experiments used Intel Skylake cores and just-in-time compilation via the popular JAX library (Frostig et al., 2018). To do this, in this section and *only* in this section, we make the following three changes to the methods (details in Appendices A.7–A.8):

• Change 1: We execute programs using the hardware's native `float32` type. As a result, intermediate values computed by the program get rounded according to IEEE rules. This produces a loss of precision throughout the program's execution that reflects the reality of practical computing. For example, rounding causes $(1 \div b) \times a$ to be less precise than $a \div b$; unlike traditional approaches, the evolutionary method can detect this difference.

• Change 2: For the accuracy objective, we use $E_{ULP}$, a maximum relative error that is normalized to *units-in-the-last-place* (*ULPs*). $E_{ULP}$ is the standard choice in numerical computing with floating-point types (in our case, `float32`). One ULP is defined as the distance to the next largest floating-point number (roughly an error of one bit). An error smaller than 1 ULP is generally considered to indicate a high-quality approximation (Muller et al., 2018).

• Change 3: For the other objective, instead of minimizing the number of operations, we maximize the throughput speed of execution of the compiled program on the hardware. This is possible because of the black-box nature of AutoNumerics-Zero (*e.g.*, the evaluation does not need to be differentiable).

BASELINES USED. The minimax approach is included because it was the top baseline in Section 4.1. That approach produces real-valued coefficients, but not every real value is representable exactly as a `float32`. This problem can be addressed with a lattice reduction method (Brisebarre & Chevillard, 2007), which we label "Polynomial/Minimax" in Figure 7. The same method does not apply to rational functions so the "Rational/Minimax" baselines round the optimized real values. Taylor expansions and Padé approximants are included too due to their popularity. To give these baselines their best chance, rational functions use only 1 division (division being the slowest operation) and all polynomials are in Horner's form, encouraging the compiler to emit fused multiply-add CPU instructions.

THE RESULTS. Figure 7 shows that top evolved programs

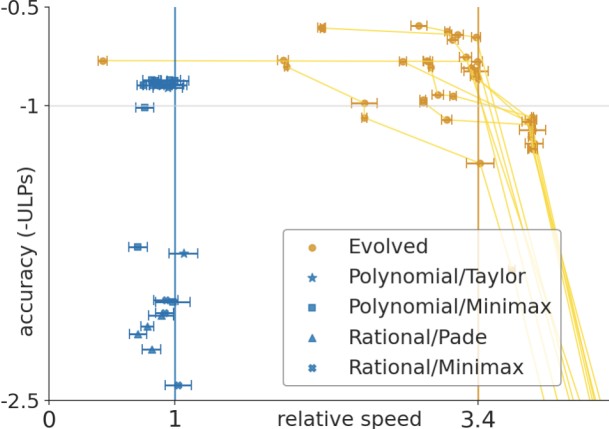

*Figure 7.* Hardware-aware approximations: evolved float-valued programs (orange) surpass the baselines (blue) by manipulating the compiler to take unusual beneficial decisions. The accuracy ($a$) measures the negative maximum relative error, measured in units-in-the-last-place. Vertical orange/blue lines highlight the speed of the fastest evolved/baseline program with $a > -1$, an accepted threshold for high-quality floating-point numerics.

are more than 3 times faster than the best baselines. This is because evolution found code that triggers an unusual decision in the compiler (details in Appendix C). Manually writing code that triggers this decision would be impractical without understanding the compiler's heuristics. In contrast, evolutionary search overcomes this obstacle without such knowledge. The novel $2^x$ program discovered (Appendix Figure 15) is almost floating-point-exact—we mathematically prove an error bound under 1 ULP in Appendix B.3.

Even though these results were evolved on a specific CPU architecture, they transfer to others. Focused on one compiler stack (Appendices A.7–A.8), Figure 8 shows that speedups are preserved across 8 years of Intel and AMD technology.

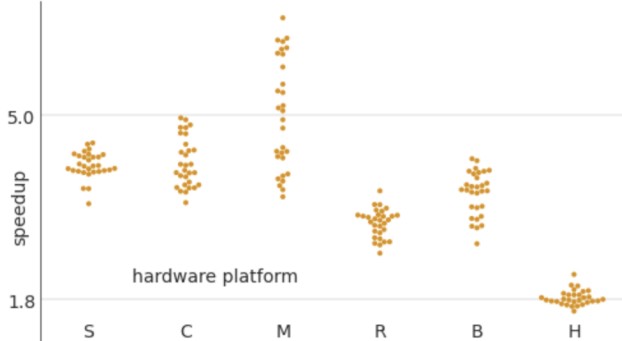

*Figure 8.* Transfer of discovered programs to other hardware platforms. Each point is an independent measurement of the speed ratio of the top evolved program to the top baseline ("top"="fastest under 1 ULP of error"). Left-to-right: Skylake (used to evolve), Cascade Lake, Milan, Rome, Broadwell, Haswell. At its worst, the evolved program was still 80% faster than the top baseline.

### 4.7. Proofs of Error Bounds

The inner loop of the evolutionary search process evaluates each candidate program only on a subset of inputs. As a result, a program discovered by the search is not guaranteed to work well for all inputs. To demonstrate that it does, we mathematically prove error bounds for our most salient evolved programs, using the following semi-automated approach. Complete proofs can be found in Appendix B. Scripts to reproduce the proofs are available online[3].

For real-valued exponential programs (Section 4.1), we prove upper bounds on the maximum relative error using interval arithmetic. As the evolved programs represent differentiable rational functions, we can iteratively subdivide the domain and prove loose error bounds on each subinterval using the IBEX library (Ninin, 2016). As the subintervals get smaller, so do their respective error bounds. The global bound is the maximum of all the subinterval bounds; with more subdivisions, the global bound becomes tighter. Intuitively speaking, we use the smallness of the subintervals to compensate for the looseness in their bounds. The results in Appendix Table 1 confirm Figure 4.

For hardware-aware programs (Section 4.6), we use a variant of interval arithmetic that accounts for floating-point lattice numerics (details in Appendix B.3). In particular, we use the *Gappa* prover to handle intermediate rounding errors under IEEE specifications (Daumas & Melquiond, 2010). Overall, this approach guarantees the forward-stability of our evolved programs (Appendix B.4). The proof bounds the error to under 1 ULP, indicating a high-quality numerical approximation (Muller et al., 2018).

### 4.8. Method Ablations

To determine the utility of the various components of AutoNumerics-Zero, we ablated each of the stages in Section 3, except the essential stage that evaluates the programs. We ablated the population selection stage by replacing NSGA-II with random search (Figure 9, left), we ablated the coefficient optimization stage by skipping CMA-ES al-

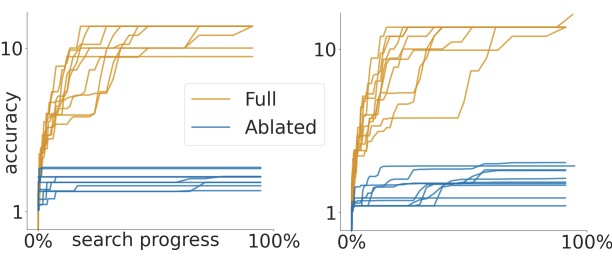

*Figure 9.* Ablations. Multiple search experiment repeats with the full method (orange lines) are compared with ablation repeats (blue lines) of the outer loop's dNSGA-II algorithm (left) and the inner loop's CMA-ES algorithm (right).

together (Figure 9, right), and we ablated the mutations by generating random graphs instead (effectively, also random search). All ablations yielded worse outcomes, confirming that each stage contributes meaningfully.

## 5. Discussion

We have demonstrated that symbolic regression can discover practical computer programs that calculate transcendental functions. This was done by comparing the results of our evolutionary search experiments to the strongest baselines we know of. The results provide evidence in favor of our hypothesis: while traditional methods are unmatched at arbitrarily high precision levels, AutoNumerics-Zero can attain practical accuracy levels with fewer operations. Here we discuss its advantages and limitations, while highlighting areas for future work.

### 5.1. Advantages

INTERMEDIATE VALUE REUSE. Representing functions as programs allowed for the discovery of code that reuses intermediate values. This was exploited by all our top evolved programs, due to the evolutionary pressure to minimize the number of operations. Figure 10 shows an example.

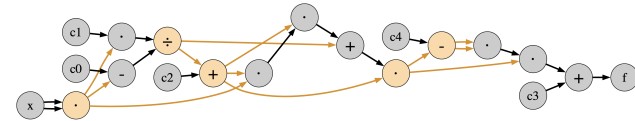

*Figure 10.* Reuse of intermediate values (orange) in the compute graph of an evolved program for $cos(x)$ computation.

NOVEL DISCOVERED FORMS. Symbolic regression searched over the space of all combinations of the basic operations $\{+, -, \times, \div\}$, which led to the discovery of novel expressions. This is starkly different from the constrained forms of traditional approaches, such as a ratio of polynomials or a continued fraction. For example, the top evolved program in Figure 5 only requires 10 operations *because* of its unconventional form:

$$f(x) = \left( \frac{c_4}{\frac{c_1}{\frac{c_3}{x}+x} + c_2 + \frac{c_3}{x} + x} - c_5 \right)^8$$

This expression is neither a ratio of polynomials nor a continued fraction—rewriting it as either would result in a higher operation count. All of our most accurate evolved programs have unconventional forms too (Appendix D). Despite this, the discovered expressions exhibit patterns. Looking across different experiments with the same conditions, we find some degree of *convergent evolution*. That is, similar features emerge because of the strong selection pressure toward

[3] https://github.com/google-deepmind/autonumerics_zero

high accuracy and few operations. For example, the best four 10-operation programs from each of the best four experiments show remarkable similarity (Appendix D). This effect is seen in natural evolution too; *e.g.*, fins evolved independently in different marine organisms because of the great swimming advantage they confer (Foote et al., 2015).

FLEXIBLE SEARCH SPACE. Another advantage we exploited is the search space's ability to incorporate instructions beyond $\{+, -, \times, \div\}$, as we showed in Section 4.4. In principle, we could also incorporate non-floating-point instructions that the CPU can execute natively, such as bit shifts or integer operations. This is not possible with traditional mathematical approaches but it is with AutoNumerics-Zero because both NSGA-II and CMA-ES are *black-box* methods. The more traditional choice of using a gradient-based method for the inner loop (Kommenda et al., 2020) would prevent the use of non-differentiable operations. The inclusion of these non-floating-point types, however, would require a typed version of the search space; we leave this to future work.

FLEXIBLE SEARCH OBJECTIVES. Evolutionary search can optimize arbitrary objectives, as we illustrated in Section 4.6, where we maximized the speed of a program on a CPU. Traditional methods are limited to simpler cost models, like the number of operations (Macon, 1955). This freedom applies to the accuracy objective too, and it can be exploited by future work to discover expressions for integrals, ordinary differential equation solutions, or inverses (*e.g.*, we can look for the inverse $f$ of a known function $g$ by minimizing the error $|1 - g \circ f|$). Finally, while in this work we have focused on accuracy (and therefore forward stability; see Appendix B.4), we speculate that future work may construct objectives to search for functions that attain backward stability too.

## 5.2. Limitations

RELIANCE ON RANGE REDUCTION. Although our search process is "zero-knowledge" on the bounded interval, extending these bounded approximations to the entire real line relies on standard range reduction. This is a shared limitation with all the baselines, as no finite combination of the basic arithmetic operations $\{+, -, \times, \div\}$ can approximate transcendental functions globally. Nevertheless, future work could aim to incorporate range reduction directly into the search process by using more recent code-discovery methods (Real et al., 2020; Chen et al., 2023; Kelly et al., 2023).

UPFRONT COMPUTE AND AMORTIZATION. The evolutionary method requires substantial upfront computational resources (summary in Section 3; details in Appendix A.9), which can pose an accessibility barrier. However, this search process constitutes a one-time cost. After the search is com-

plete, discovered programs can be freely shared and reused, amortizing the cost. This amortization could prove particularly significant in high-performance computing applications bottlenecked by trillions of transcendental function evaluations (*e.g.*, Wu et al., 2023). Moreover, we view our specific search process as a starting point, drawing the following historical parallel to neural architecture search (NAS). Early NAS relied on compute-intensive reinforcement learning and evolutionary search to establish viability. That initial proof of capability inspired highly efficient algorithms like DARTS (Liu et al., 2019). We anticipate a similar trajectory here: demonstrating that symbolic regression can discover state-of-the-art programs is an initial step, which we expect will motivate future algorithmic improvements to lower resource requirements.

INTERPRETABILITY–QUALITY TRADEOFF. Traditional approximation methods are interpretable because they conform to rigid, hand-designed forms. For example, a truncated Taylor expansion is constrained to a polynomial form that can be interpreted as a sequence of increasingly smaller corrections. AutoNumerics-Zero discards these structural priors to search over unconstrained forms resulting from arbitrary combinations of the basic operations. It is this unconstrained search that enables the discovery of novel programs with significantly improved accuracy for given operation counts. This shift is somewhat analogous to the broader machine learning transition from hand-crafted features to deep learning, where explicit structural priors are traded for empirical performance. In practice, the specific application should determine the optimal balance along this interpretability–quality tradeoff.

## 5.3. Conclusion

In this work, we introduced AutoNumerics-Zero, a zero-knowledge symbolic regression method that automatically discovers programs that compute transcendental functions. AutoNumerics-Zero shifts the paradigm away from the traditional emphasis on manually developing expressions that are capable of arbitrary accuracy. Instead, we showed that by targeting high-but-finite accuracy, novel expressions emerge that can outperform established baselines while maintaining mathematically proven error bounds. Future work can attempt to unlock additional gains by taking further advantage of the black-box nature of our search process: for example, by including the range reduction step directly into the search or by adding hardware-native operations, such as bit shifts, into the operation set. Ultimately, this work serves as a rigorous demonstration that evolutionary search offers a new path to function approximation that takes into account the intricacies of digital computing.

## Acknowledgements

**We would like to thank**: Daniel S. Fisher, Michael Brenner, Blaise Aguera y Arcas, Ben Laurie, Thomas Fischbacher, Rif A. Saurus, Matt Hoffman, and Kalyanmoy Deb for useful discussions; Sameer Agarwal and Rasmus Larsen for advice on numerics; Yingtao Tian, Yujin Tang, Yingjie Miao, and Daiyi Peng for code contributions; and more generally the broader Google Research, Google Paradigms of Intelligence, and Google DeepMind teams.

**Author contributions**: ER led the project, mentoring MR, CdS, MG, YC, AV, and DHP. ER developed the methods, constructed baselines, and ran experiments. ER, MR, MG, AV, and DHP contributed code. CdS and YC provided low-level programming expertise and assembly code analysis. MF contributed the error bound proofs. MR, CdS, and MG open-sourced the code. ER wrote the paper. EDK and QVL edited the paper and provided organizational support.

## Impact Statement

This paper presents work whose goal is to advance the field of Machine Learning. There are many potential societal consequences of our work, none of which we feel must be specifically highlighted here.

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

# Appendix

## A. Method Details and Clarifications

This section elaborates on the Methods Outline presented in the main paper (Section 3); some aspects of the methods are presented there but not here.

### A.1. Target Functions and Their Domains

The main objective is to discover programs that compute a given *target function* $g(x)$. For the target function $g(x) = 2^x$, we searched for an expression in the $(0, 1]$ interval. This incurs no loss of generality because any input on the real line can be quickly transformed to a corresponding input in $(0, 1]$ using the standard process of *range reduction* (Appendix Method 2). Both our method and all the baselines rely on range reduction for the reasons explained in the setup of Section 4.1.

---

**Method 2** Standard Range Reduction for $g(x) = 2^x$

---

**input** a floating-point value $x \in (-\infty, \infty)$.
**Require:** a function $\tilde{g}(u)$ s.t. $\tilde{g}(u) = 2^x$ for $x \in (0, 1]$.
**Require:** a function $\hat{g}(k)$ s.t. $\hat{g}(k) = 2^k$ for $k \in \mathbb{Z}$.
**output** the value $g(x)$.
   $\eta = \lceil x \rceil - 1$   # $\implies \eta \in \mathbb{Z}$
   $\xi = x - \eta$   # $\implies \xi \in (0, 1]$
   **return** $\hat{g}(\eta)\tilde{g}(\xi)$

---

In addition to $g(x) = 2^x$, we also considered the following functions:
- $g(x) = \cos(x)$ on $[0, \pi/2]$,
- $g(x) = \mathrm{erf}(x) = \frac{2}{\sqrt{\pi}} \int_0^x e^{-t^2}\, dt$ on $(0, 2]$;
- $g(x) = \mathrm{Ai}(-7\,x)$ on $[0, 1]$; and
- $g(x) = I_{1/2}(x)$ on $(0, 1]$,

where $Ai$ denotes the Airy function of the first kind and $I_{1/2}(x)$ denotes the modified Bessel function.

### A.2. Search Space Details

The search space was described in Sections 1 and 3. Some additional details follow.

We limit the number of instructions in the program to 100 in order to control execution time and memory use, though this is not a fundamental limitation. The operations available were chosen to be $\{+, -, \times, \div\}$ so that results can be easily compared with approximations produced by the baselines.

In compute graph figures, the input is denoted by a vertex labeled $x$ and the output by a vertex labeled $f$. Additional optional input vertices, denoted $c_i$, represent the *coefficients*.

Internal vertices represent instructions encoding the mathematical operations. For binary operations, the order of arguments to an operation is determined by the top-to-bottom order of a vertex's in-edges.

Vertices are assigned by the search process a fixed random integer *ordering parameter* (not shown in compute graph figures, but implicit in the code version), which resolves any ambiguities in the ordering of instructions, subordinate to the topological order of the compute graph.

### A.3. Selection Stage Details

The outer loop of the search method, including the selection stage, was described in Section 3; more details follow.

The NSGA-II algorithm (Deb et al., 2002) maintains a population of P candidates, where P is the *population size* hyperparameter. In our case, the candidates are the computer programs. For each parent program, the mutations produce 2 P child programs. NSGA-II selects new parents from among these children by considering the two objectives of the search: the program's accuracy ($a$) and number of operations ($n$) (or speed, in the case of the hardware-aware section). To do this, NSGA-II classifies the children into *fronts* in a process called *non-dominated sorting*. These fronts are disjoint subsets of the population with the property that elements within a front cannot dominate each other by this definition: program $p_1$ dominates $p_2$ iff $p_1$ is better than $p_2$ at one objective and no worse at the other; that is,

$$(a_{(p_1)} \geq a_{(p_2)} \land n_{(p_1)} \leq n_{(p_2)}) \land$$
$$(a_{(p_1)} > a_{(p_2)} \lor n_{(p_1)} < n_{(p_2)})$$

The first front contains all the programs not dominated by any other and is therefore the Pareto front of the population. The next front contains all the remaining programs not dominated by any other remaining program, and so on. The fronts are therefore ordered. NSGA-II will follow this order to select the top $P$ programs as the next population. NSGA-II therefore must maintain a centralized population of size $P$; the computational complexity of the selection process described is $O(P^2)$. We are interested in a distributed setting. To decentralize the search process (see Section 3), dNSGA-II recognizes that non-dominated sorting works even on small samples of the population. These samples have sizes much smaller than P. The outline of the process is shown in Method 1. First, each worker receives a sample of 2 S programs from other workers. It then carries out the selection resulting in S parents. These are mutated to

produce 2 S children, which are emitted for other workers to use. The sample is therefore treated analogously to a tournament in tournament selection approaches (Goldberg & Deb, 1991).

### A.4. Mutation Stage Details

The mutations were described in Section 3 and illustrated in Figure 3; more details follow. There is an equal probability of making one of the three mutations in the figure or doing nothing. When making a change, there is an equal probability of mutating the vertices (mutations 1 or 2 in Section 3) or mutating an edge (mutation 3); when mutating a vertex, the probability of inserting (mutation 1) is half the probability of deleting (mutation 2) so as to regularize the size of the programs. When inserting a new vertex, the operation performed is randomly chosen from the four vertex operations $\{+, -, \times, \div\}$ or the insertion of a positively or negatively initialized coefficient, with uniform probability. Even though Figure 3 does not show it in the examples, the insert and delete mutations can also operate on the coefficient vertices (not just the operation vertices). After applying a mutation to the compute graph, the graph is pruned to remove vertices that are no longer in the path from input ($x$) to output ($f$). We did not tune any of these probabilities.

### A.5. Coefficient Optimization Stage Details

The coefficient optimization stage was described in Section 3; more details follow.

We are interested in maximizing the programs' accuracy. This accuracy is defined in relation to the maximum relative error over the domain $\mathcal{D}$ of the target function. More precisely, the accuracy $a$ of an approximation $f$ of a target function $g$ is:

$$a = -\log_{10} \max_{x \in \mathcal{D}} \frac{|g(x) - f(x)|}{|g(x)|}$$

Note that Sections 4.2–4.5 used the absolute error instead of the relative error for the reasons described therein; that is:

$$a = -\log_{10} \max_{x \in \mathcal{D}} |g(x) - f(x)|$$

Since $\mathcal{D}$ is infinite, for the purposes of coefficient optimization, we estimate the accuracy by taking the max only over a finite random subset $\mathcal{T}$.

We used the CMA-ES method to optimize the coefficients. CMA-ES is a popular evolutionary algorithm used for continuous optimization; it keeps a population of candidate solutions which it improves iteratively by replacing it with a new population. Each candidate is a vector of values, which in our case are the coefficients. In outline, the new population is formed by repeatedly sampling from a Gaussian

distribution with the mean and covariance of the best half of the previous population (Hansen & Ostermeier, 2001).

*Further details: $\mathcal{T}$ contains $10^3$ values chosen uniformly randomly from the domains described in Appendix A.1. Coefficients are initialized as $\pm 10^{-\alpha}$ where $\alpha$ is uniformly distributed in $[0, 8]$; the coefficient's sign has an equal probability of being positive or negative. CMA-ES uses a population size of $128$, $10^4$ generations, and the following early-stopping strategy: after the first $100$ generations, if the latest half of the generations does not result in any improvement in the maximum of the population, the optimization is stopped early.*

### A.6. Evaluation Stage Details

The evaluation stage was described in Section 3; more details follow. We measure the accuracy as before, only that this time we take the max over a larger subset $\mathcal{V}$ of $\mathcal{D}$, roughly 10 times larger than $\mathcal{T}$. These examples were not seen during optimization, which helps reduce overfitting.

### A.7. Coefficient Optimization Details (Hardware Aware)

Results Section 4.6 used a modified version of the coefficient optimization stage, which was outlined there; more details follow.

The optimization stage translates the compute graph produced by the mutations into a standard programming language by converting each of its vertices into JAX-NumPy operations (`jnp.sum`, `jnp.multiply`, *etc.*). The result is compiled with XLA (Sabne, 2020) and its coefficients are optimized.

We are interested in maximizing the programs' accuracy. Again, the accuracy is defined in relation to the maximum relative error. However, since we are now dealing with floating-point types instead of real numbers, it is standard to express the error in terms of units-in-the-last-place (ULPs), as was described in the main text. More precisely, the accuracy of an approximation $f$ of a target function $g$ is:

$$a = -\max_{x \in \mathcal{F}} \frac{|g(x) - f(x)|}{\text{ulp}(g(x))}$$

where $\mathcal{F}$ denotes the set of all `float32` numbers and $\text{ulp}(y)$ denotes one ULP of $y$. An ULP is defined as the distance to the closest larger value representable as a floating-point number. Since the standard threshold for high-quality numerics is 1 ULP, we are interested in the behavior of speed near the $a = -1$ boundary; this is the reason we do not use a logarithmic scale to compute $a$ (*cf*. Appendix A.5). Since $\mathcal{F}$ is very large, for the purposes of coefficient optimization, we estimate the accuracy by taking the max over a smaller random subset $\mathcal{T}$.

Optimization is done with CMA-ES as in Appendix A.5.

*Further details: The JAX version used is 0.4.11; the Jaxlib version is 0.4.11. All the lower level compilation is done by the*

*XLA and LLVM included in Jaxlib. During the search process, JAX NumPy flushes subnormal numbers to zero; this could in principle affect results, but our testing of selected evolved functions accounts for this (Appendix A.10). For speed measurements, the vector of inputs has size 10000, the stack depth is 100, and the number of repeats is 1000. The CMA-ES hyper-parameters are the same as in Appendix A.5.*

## A.8. Evaluation Stage Details (Hardware Aware)

Results Section 4.6 used a modified version of the evaluation stage, which was outlined there; more details follow.

The evaluation stage binds the coefficients obtained from the optimization stage, re-compiles the program, and measures its accuracy and speed. Because this evaluation process needs to be embedded in the outer loop of the symbolic regression process, we use Python/JAX as the programming language, which allows programmatically calling the compilation of arbitrary functions through its just-in-time (JIT) mechanism. This is a choice of convenience, not necessity.

In the evaluation phase, we measure the error again, but on a set $\mathcal{V}$ with unseen examples, roughly 10 times as many as were used in $\mathcal{T}$. This helps reduce optimization over-fitting. During evaluation, we also measure the *speed* of multiple evaluations in an embarrassingly parallel regime, optimizing for throughput. Some precautions are necessary to get accurate timings. First, overheads must be sufficiently reduced, so instead of compiling a single execution of the $f(x)$ program, we compile it to apply to a vector of all the numbers in $\mathcal{V}$. This is not enough, however, and so each input is acted upon by a stack of the form $f \circ h \circ f \circ h \ldots$, where $h$ is a clipping function to keep the values in range. Stacking ensures the optimizer does not compile away unused results. Finally, we must guard against interruptions by concurrent processes, as we are operating in a distributed environment with little control over task scheduling. For this, we perform many repeats of the timing measurement. Any measurement that is interrupted or must share the CPU should result in an overestimate of the timing. Thus, of all the repeats, the tightest approximation is the shortest measurement. A similar strategy is used in Python's `timeit` module.

## A.9. Resources

The search process is distributed on 100–10k processes for 1–4 days on a custom cluster of Skylake cores, with each process using up to 1 core. The most compute-intensive experiments were those in Section 4.6; in that case, reaching the best program required 9.6k core-days. Our main results, in Section 4.1, used 6k core-days; however, this is likely to be reduced significantly by relaxing our self-imposed requirement that the all experiments should use the same search process configuration with the same hyper-parameters.

## A.10. Testing Evolved Programs Against Baselines

Once the evolutionary search process is complete, we retrieve programs in the Pareto front of the final population and thoroughly assess their accuracy and, where applicable, their speed. We compare them against baselines, chosen from the following:

- **Polynomial/Taylor**: Horner-form Taylor expansions of order $M$ (see below) about the middle of the interval (unless otherwise noted).
- **Rational/Padé**: Padé approximants of order $(M,M)$ about the middle of the interval (unless otherwise noted), with Horner-form numerator and denominator.
- **Chebyshev**: Polynomial approximation of order $M$ due to Chebyshev (Tchébychev, 1858), where the coefficients were computed with numerical integration.
- **C. Frac. / Euler**: continued fractions due to Euler (1748).
- **C. Frac. / Gauss**: cont. fractions due to Gauss (1813).
- **C. Frac. / Macon**: cont. fractions due to Macon (1955).
- **Polynomial/Minimax**: Horner-form polynomials of order $M$. For the hardware-aware cases, the coefficients were optimized with the Sollya library (Chevillard et al., 2010) by the lattice reduction method in Brisebarre & Chevillard (2007). For all other cases, they were optimized in the same manner as the Rational/Minimax case below.
- **Rational/Minimax**: Ratios of order-$M$ Horner-form polynomials, with coefficients optimized with the *Mathematica* software suite (Wolfram Research, Inc., 2023) to minimize the maximum relative error. Mathematica carries out this optimization by first approximating the coefficients by fitting the rational function to a few points of the target function and then improving them with the Remez algorithm (Remez, 1969).

These choices have the goal of giving the baselines their best chance. For each case, we use baselines with various $M$ values, covering a wide range of accuracies (in the case of hardware-aware baselines, we increase $M$ until the results plateau).

To compare the accuracy of programs and baselines, we use a tiered testing approach, placing more rigor on the verification of our most salient results while relying on tight estimates for the rest. For the best evolved exponential programs (Section 4.1 and Figure 4), we prove upper bounds on the error (Appendix B.2). For the top hardware-aware program (Section 4.6), we proved a 1-ULP bound on the error (Appendix B.3). For most bulk figure comparisons between evolved results and baselines, we measured the error over a test set $\mathcal{U}$ that is 100 times larger than the evaluation set $\mathcal{V}$; these should be reasonably tight estimates because the functions are well behaved.

To test the speed of programs and baselines in the hardware-

aware section, we repeat 10 times the speed measurement described in Appendix A.8. Relative speeds are measured with respect to the top baseline, where "top" is the fastest with less than 1 ULP of error.

_____ *Details: To compute coefficients for the Rational/Minimax exponential baselines, we used the Mathematica command:*
```
MiniMaxApproximation[2^x, x, 0, 1, M, M,
WorkingPrecision -> 100]
```
*We proceeded similarly for other minimax cases, except for the hardware-aware float-valued Polynomial/Minimax exponential baseline because the lattice-reduction method needs a specialized library, so we used the Sollya command:*
```
poly = fpminimax(2^x,8,[|24...|],[0;1],relative);
```

## B. Proofs of Error Bounds: Details

The inner loop of the evolutionary search process evaluates each candidate program (*i.e.* function) on a subset of inputs. As a result, a program discovered by the search is not guaranteed to work well for all inputs. To obtain an accuracy guarantee, it is necessary to prove an error bound for the program. In this section, we demonstrate mathematically the correctness of the main programs evolved by our method.

### B.1. Formalities

Section 4.1 presented real-valued evolved programs that approximate the exponential function $2^x$. Their full code is given in Appendix D. In this section, we make some formal clarifications that the proofs will rely upon.

**Well-defined functions.** Here we check that the discovered functions are well defined in $(0,1]$. This is not guaranteed by the evolutionary process that discovered them, as it only evaluates the programs at a finite number of points. Because functions are composed of the $\{+, -, \times, \div\}$ operations, they are defined almost everywhere. In principle, however, they may not be defined at isolated points corresponding to the zeros of the denominators of the $\div$ operations. Nevertheless, this is not a problem in our case because such zeros lie outside $(0,1]$. To prove this, for each discovered program, we parse its compute graph in topological order ("from input to output"), ensuring that all denominators we encounter have no roots in $(0,1]$. To ensure that we have no roots in $(0,1]$, we can employ one of two different methods: (1) One method is to treat the constants $c_i$ as symbols. If the polynomial which corresponds to the denominators factors into polynomials of degree less than five, we can symbolically find the roots. Then by plugging in the values for the $c_i$, we can determine them numerically and verify that they lie outside of $(0,1]$. This evaluation can be made numerically rigorous by employing interval arithmetic methods, similar to those in Appendix B.2. (2) If, on the other hand, the denominator polynomials are of higher degree, we must resort to the following more general method. We

can rigorously compute bounds on all the roots to show they are outside $(0,1]$; we do this by using a combination of homotopy continuation methods and interval arithmetic. For this purpose, we use the HomotopyContinuation.jl package, written in the Julia Programming Language; see Breiding & Timme (2018). This way, we show inductively that each denominator is a well-defined rational function on $(0,1]$.

For all the discovered functions and at all places where denominators appear in the compute graph, the degrees of the polynomials are low enough that method (1) in the previous paragraph can be used. Nevertheless, we also run method (2).

For example, in the case of the discovered 10-operation exponential program (Figure 5), the function computed was:

$$f(x) = \left( \frac{c_4}{\frac{c_1}{\frac{c_3}{x}+x} + c_2 + \frac{c_3}{x} + x} - c_5 \right)^8$$

Parsing its compute graph in topological order yields the following denominators: $x$ (from $\frac{c_3}{x}$), $\frac{c_3}{x} + x$, and $\frac{c_1}{\frac{c_3}{x}+x} + c_2 + \frac{c_3}{x} + x$. By substituting the $c_i$, we can estimate the roots to be: the single root 0, the two roots $\pm 76.0495 i$, and the four roots $50.8891 \pm 56.5139\,i$ and $74.3871 \pm 15.8143\,i$, all outside of $(0,1]$.

**Closed support.** All of the discovered functions can be continuously extended from $(0,1]$ to $[0,1]$. For example, in the case of the 10-operation program (Figure 5), the singularity at $x = 0$ is removable by defining $f(0) = c_5{}^8$. Thus, the functions have the closed support needed for the interval arithmetic arguments used in the proofs below.

**Differentiability.** As shown above, discovered functions can be extended to *rational* functions in $[0,1]$ with no poles. Thus, they are differentiable in $[0,1]$.

### B.2. Real-valued Error Bounds

Section 4.1 presented real-valued evolved programs that approximate the exponential function $2^x$. Their full code is given in Appendix D. In this section, we prove upper bounds on their real-valued maximum relative error over the entire real line. Again, because of the range reduction method (Appendix A.1), without loss of generality, we can focus on the $[0,1]$ interval. Note that Appendix B.1 has already shown that the functions are differentiable in $[0,1]$.

As the evolved programs represent differentiable functions, we can use interval arithmetic techniques to automatically construct the proofs. We chose the IBEX library to do the necessary calculations (Ninin, 2016). In outline, we construct a proof by iteratively splitting the $(0,1]$ interval and proving loose error bounds on subintervals. As the subintervals get smaller, so do their respective error bounds. The global bound is the maximum of all the subinterval

bounds; thus, with more subdivisions, the global bound becomes tighter. Intuitively speaking, we use the smallness of the subintervals to compensate for the looseness in their bounds.

In more detail, consider an evolved function $f\colon (0,1] \to \mathbb{R}$ for which we want to prove an error bound. Let $\mathcal{E}\colon (0,1] \to \mathbb{R}$ be the relative error function w.r.t. the true exponential:

$$\mathcal{E}(x) = \frac{f(x) - 2^x}{2^x}$$

We wish to prove a tight error bound $\epsilon$ on $\mathcal{E}$ over $(0,1]$.

For a given subinterval $[a,b] \subseteq (0,1]$, we wish to establish a loose bound $\eta([a,b])$ for $f$ over $[a,b]$. To do this, we first establish a Lipschitz bound $L$ on the derivative, such that for all $x \in [a,b]$ we have $|f'(x)| \le L$. This can be done by using interval arithmetic techniques:

$$L = \mathcal{UB}\left(\left|f'([a,b])\right|\right) \tag{1}$$

where $\mathcal{UB}$ denotes the upper bound. We highlight that in Equation 1, all the operations are of the interval arithmetic kind; that is, given an interval $I$, the expression $g(I)$ represents a new interval $J$ satisfying $J \supseteq g(s) \mid \forall s \in I$. Interval arithmetic therefore gives us a method for *computing* $L$ from $f$. More details can be found in Tucker (2011).

Given the bound $L$ on the derivative, we can apply the mean value theorem to deduce that for all $x,y$ in $[a,b]$ there exists a $c$, such that

$$|f(x) - f(y)| = |f'(c)| \cdot |x - y| \le L|x - y|$$

In particular, let $m$ be the midpoint: $m = \frac{a+b}{2}$. The above implies that for all $x$ in $[a,b]$:

$$|f(x) - f(m)| \le \frac{L|b-a|}{2}$$

This establishes that $f(x)$ cannot deviate significantly from $f(m)$ as the interval $[a,b]$ becomes small. In fact, $f(x)$ must lie within the interval $f([x,x]) + [-L,L] \cdot (b-a)/2$. In other words, using interval arithmetic notation:

$$\eta([a,b]) = \mathcal{UB}\left(\left|f([x,x]) + [-L,L] \cdot \frac{b-a}{2}\right|\right)$$

To prove a given error bound $\epsilon$ over the entire $(0,1]$ interval, we proceed as follows. First we calculate the upper bound $\eta$ with the method described above. If $\eta \le \epsilon$, we are done. If $\eta > \epsilon$, we subdivide the interval into two equal parts and apply the method recursively. If the method terminates, we have proven the error bound $\epsilon$.

Using the method just described, we prove the bounds listed in Appendix Table 1, confirming the results of Figure 4 in the main text.

| Function | Error Bound |
|---|---|
| f2 | 0.0415 |
| f3 | 0.00123 |
| f4 | 0.0003072 |
| f5 | $6.372 \times 10^{-6}$ |
| f6 | $4.016 \times 10^{-7}$ |
| f7 | $8.417 \times 10^{-10}$ |
| f8 | $1.360 \times 10^{-11}$ |
| f9 | $2.15 \times 10^{-13}$ |
| f10 | $5.40 \times 10^{-15}$ |

*Table 1.* Proven error bounds on evolved real-valued $2^x$ approximations. Each line corresponds to one of the programs in Appendix Figure 11 and to an evolved point in Figure 4.

### B.3. Hardware-aware (float-valued) Error Bounds

Section 4.6 presented a float-valued program that approximates the exponential function $2^x$ with less than 1 ULP of error over the set of all `float32` numbers. Here we prove this error bound rigorously. As stated before, because of range reduction (Appendix A.1), without loss of generality, we can focus on the floating-point numbers in $(0,1]$, a set which we will denote as $\mathbb{F}$. The rounding due to floating-point operations demands a different proof from the one used above for real-valued functions.

We want to show that the function $g(x) = 2^x$ can be approximated by $f(x)$, our discovered function, over the set $\mathbb{F}$, with a maximum relative error $\mathcal{E}$ of less than 1 ULP. Since $ulp(2^x) \ge 1$ for $x > 0$, the relative error can be bounded by:

$$\mathcal{E} = \frac{|2^x - f(x)|}{\mathrm{ulp}(2^x)} \le \frac{\tilde{\mathcal{E}}}{\mathrm{ulp}(1)} \tag{2}$$

where $\tilde{\mathcal{E}}$ denotes the absolute error $|2^x - f(x)|$.

The strategy will be to define auxiliary functions $h_{\mathbb{R}}(x)$ and $h_{\mathbb{F}}(x)$ and split the absolute error into three parts: (A) the error $\tilde{\mathcal{E}}_A$ of approximating $g$ with $h_{\mathbb{R}}$, (B) the error $\tilde{\mathcal{E}}_B$ of approximating $h_{\mathbb{R}}$ with $h_{\mathbb{F}}$, and (C) the error $\tilde{\mathcal{E}}_C$ of approximating $h_{\mathbb{F}}$ with $f$, so that:

$$\tilde{\mathcal{E}} \le \tilde{\mathcal{E}}_A + \tilde{\mathcal{E}}_B + \tilde{\mathcal{E}}_C \tag{3}$$

Part (A) has the intuitive goal of removing the difficulties associated with transcendental functions from the rest of the proof. Define $h_{\mathbb{R}}(x)$ as the 20th order Taylor expansion of $g$ about zero, computed with real-valued operations. That is, even though the domain of $h_{\mathbb{R}}(x)$ is still the floating-point subset $\mathbb{F}$, the additions and multiplications are real-valued and may therefore produce numbers outside of $\mathbb{F}$; thus, $h_{\mathbb{R}}(x)$ is real-valued. $\mathcal{E}_A$ can therefore be bounded by Taylor's theorem; the Lagrange form of the Taylor remainder

gives:

$$\tilde{\mathcal{E}}_A \leq \max_{x,y \in \mathbb{R}} \left| \frac{f^{(21)}(x)\, y^{21}}{21!} \right| \leq \frac{2(\ln 2)^{21}}{21!} \leq 10^{-16}$$

Part (B) transitions from real-valued operations to floating-point operations. Define $h_{\mathbb{F}}$ also as the 20th order Taylor expansion of $g$ about zero, except that this time we use floating-point multiplications and additions. In other words, after every intermediate operation, the result is rounded to the nearest float, reflecting actual computer hardware calculation. Because of this rounding, we need to specify that the polynomial is written in Horner's form: $h_{\mathbb{F}}(x) = a_0 + x(a_1 + x(...))$, where the $a_i$ are the Taylor coefficients (rounded to the nearest float). We can now bound $\tilde{\mathcal{E}}_A$ by using the automated prover *Gappa* (Daumas & Melquiond, 2010; Muller et al., 2018). In order to represent $h_{\mathbb{R}}(x)$ without losing accuracy, we encode it in Horner's form, with the $nth$ coefficient represented as $\frac{(\log 2)^n}{n!}$ with log2 as a constant in the interval $[0.693147180559945309417, 0.693147180559945309418]$. For example, a 3rd order Taylor expansion would be encoded in our *Gappa* program as

$$1 + x\left(\log 2 + x\left(\frac{(\log 2)^2}{2} + x\frac{(\log 2)^3}{6}\right)\right)$$

We encode $h_{\mathbb{F}}$ in the *Gappa* program in the same way as in the optimized program and take advantage of *Gappa*'s support for $x86\_80$ accuracy with IEEE-754 rounding. The prover shows that $\tilde{\mathcal{E}}_B \leq 10^{-18}$.

Part (C) can now complete the proof by bounding $\tilde{\mathcal{E}}_C$ using only floating-point additions and multiplications over $\mathbb{F}$. This bound can be calculated exactly by exhaustive evaluation with a computer. We find that $\tilde{\mathcal{E}}_C \leq 7.8 \times 10^{-8}$.

Putting everything together, Equations 2 and 3 imply that $\mathcal{E} \leq 0.654$.

### B.4. Stability of Discovered Programs

The forward stability of our real-valued and float-valued exponential programs is guaranteed by the foregoing proofs, as follows. We first consider the more complex case of the float-valued proofs, then the real-valued proofs, and finally we briefly comment on backward stability.

Our float-valued proofs in Section B.3 guarantee forward stability, as they provide a uniform bound on the end-to-end relative error while accounting for the maximum possible floating-point rounding error at each intermediate step. This is possible because our method produces explicit, finite computer programs (on the reduced interval). The main tool we used, the Gappa prover, was specifically designed to perform forward error analysis over floating-point arithmetic

(Daumas & Melquiond, 2010). A critical subtlety of forward error analysis (and a common source of instability) is the potential for errors to accumulate at intermediate steps due to the loss of precision from floating-point rounding. The Gappa prover directly accounts for this, as it simulates IEEE-754 rounding at every intermediate calculation, compounding the worst-case rounding error at each operation in the program's compute graph, and propagating these bounds forward to the output node. Furthermore, regarding mathematical correctness, a program could theoretically perform error-free intermediate calculations yet arrive at the wrong answer due to a poor mathematical approximation; our proofs preclude this by strictly bounding the total error against the true target function. Using Gappa avoids analyzing the formulas purely algebraically, which would quickly become cumbersome. Thus, this end-to-end bound guarantees both the stability of the approximation under finite-precision arithmetic and its mathematical correctness. Beyond the reduced interval, stable range reduction methods exist independently of our approach (*e.g.*, Brisebarre et al., 2005).

While the main subtlety of forward stability lies in the finite-precision arithmetic described above, our real-valued proofs in Section B.2 also guarantee forward stability by mathematically proving uniform error bounds. Because real-valued expressions do not suffer from intermediate floating-point rounding errors, their forward stability is a more direct consequence of the established relative error bounds, and can be justified with a similar but simpler argument to the float-valued case above.

Focusing on *forward* stability is a natural choice because our evolutionary process explicitly optimizes for accuracy. However, backward stability would provide another useful point of view. We suspect that the objectives of our evolutionary process could be adjusted to search for backward-stable programs; we leave this exploration to future work.

## C. Speed of Hardware-Aware Programs

In Section 4.6, we outlined the reason why the top evolved program is faster than all the baselines by a significant amount; here we provide additional detail. The cause of the relative speedups shown in Figure 7 is that the compiled baseline functions are routed by the compiler through a *parallel task assigner* dispatch function before reaching the actual computation of the exponential function. This adds significant overhead including function calls to the otherwise-simple exponential computation, dramatically reducing performance. In the version of the XLA compiler used here, the reason for this routing is a decision made in the *CPU Parallel Task Assigner* pass of the High Level Optimizer (HLO), which determines whether or not this dispatcher is added for each *fused computation* present in

the function prior to this optimization pass. Fused computations are groups of operations that the compiler believes can be executed efficiently together in common loops (The XLA Team, 2017). The decision is based primarily on the number of operations the fused computation contains in the HLO Intermediate Representation (IR) (The XLA Authors, 2023b). It also considers the number of bytes the fused computation processes, but that number is constant for all the exponential function implementations that we evaluated. The grouping of HLO operations into fused computations is handled by the prior *Fusion* HLO pass, which decides whether groups of operations are fusible or not based on a series of heuristics (The XLA Authors, 2023a). Examining the intermediate HLO IR immediately prior to the CPU Parallel Task Assigner shows that the baseline has a single, long fused computation, but the evolved function's operations are split into two separate fused computations. Since the CPU Parallel Task Assigner pass operates at the fused computation level and not the full function level, it appears that the evolved function's computations have fewer operations (each), leading to the conclusion that the function is I/O-bound and should not be compiled for parallelization. Without the need for parallelization, the calls to the evolved function are not routed through the *parallel task assigner* dispatch function. With the baseline's longer fused computation, the conclusion is that the function is CPU-bound and therefore should be compiled with routing through the dispatch function. This behavior of performance being dramatically different according to complex compiler heuristics is difficult to predict when writing high-level code; on the other hand, the evolutionary process can optimize it without requiring any specific understanding of these issues.

## D. Discovered Programs

Appendix Figure 11, in the next page, contains all the evolved programs referred to in Figure 4 (Section 4.1). We mentioned in Section 5 that the results show evidence of convergent evolution. For example, the following are the four best 10-operation programs from four different experiments; they show remarkable similarities in the discovered code (the constants have been grossly approximated for ease of readability):

$$f(x) \sim \left( \frac{500}{-250 + x + \frac{5700}{x} + \frac{15000x}{5700 + x^2}} + 1 \right)^8$$

$$f(x) \sim \left( \frac{690}{346 + \frac{7800}{x} - \frac{40000}{\frac{-7800}{x} - x}} + 1 \right)^8$$

$$f(x) \sim \left( \frac{-690}{-350 + \frac{-7800}{x} + \frac{40000}{\frac{-7900}{x} - x}} + 1 \right)^8$$

$$f(x) \sim \left( \frac{-0.17}{0.087 - 0.00069x - \frac{1}{x} + \frac{0.0018}{\frac{1}{x} + 0.00069x}} + 1 \right)^4$$

Also, we consistently observe intermediate-value reuse (top program in 10 out of 10 experiments) and the use of a single division in the compiled program (top program in 9 of 10 experiments).

Appendix Figure 12, in the next page, shows the evolved programs represented in Figure 6 (left). Appendix Figure 13 shows additional programs referred to, but not shown, in Section 4. Appendix Figure 14 shows the evolved programs represented in Figure 6 (right). Appendix Figure 15 shows the best program from Section 4.6.

```
def f2(x):
    c1 = -2.1258595374472384
    c2 = -2.0413845597733418
    x1 = c2 + x
    x2 = c1 / x1
    return x2
```

```
def f3(x):
    c1 = -8.387819235563974
    c2 = 1.4427239805682266
    c3 = -3.4355225277901402
    x1 = c3 + x
    x2 = c1 / x1
    x3 = x2 - c2
    return x3
```

```
def f4(x):
    c1 = -13.889185227358549
    c2 = -6.3103598040432605
    x1 = c2 + x
    c3 = 1.2011665727304095
    x2 = c1 / x1
    x3 = x2 - c3
    x4 = x3 * x3
    return x4
```

```
def f5(x):
    c1 = 25.596749740144819
    c2 = 17.746150088734609
    x1 = c1 / x
    c3 = -0.99999366143081858
    c4 = -8.8505996518073289
    x2 = x + x1
    x3 = c4 + x2
    x4 = c2 / x3
    x5 = x4 - c3
    return x5
```

```
def f6(x):
    c1 = 34.839796464204852
    c2 = -17.414314151974395
    c3 = 100.52501140663516
    x1 = c3 / x
    x2 = x + x1
    x3 = c2 + x2
    x4 = c1 / x3
    c4 = -0.99999980025884683
    x5 = x4 - c4
    x6 = x5 * x5
    return x6
```

```
def f7(x):
    c0 = -31.36503547149854
    c1 = 237.4188094069423
    c2 = 62.73011283808836
    c3 = -0.9999999991613342
    c4 = 90.500418999822784
    x1 = c4 / x
    x2 = x + x1
    x3 = c1 / x2
    x4 = x3 + x2
    x5 = c0 + x4
    x6 = c2 / x5
    x7 = x6 - c3
    return x7
```

```
def f8(x):
    c1 = 947.16279416218413
    c2 = -62.660057097633249
    c3 = 361.59742194279426
    x1 = c3 / x
    x2 = x + x1
    c4 = 125.32011684077629
    c5 = -0.99999999999334788
    x3 = c1 / x2
    x4 = x3 + x2
    x5 = c2 + x4
    x6 = c4 / x5
    x7 = x6 - c5
    x8 = x7 * x7
    return x8
```

```
def f9(x):
    c1 = -125.28498998901401
    c2 = 3786.1251186399709
    c3 = -250.56998014383566
    c4 = 1445.9842709817003
    c5 = 0.99999999999994771
    x1 = c4 / x
    x2 = x + x1
    x3 = c2 / x2
    x4 = x3 + x2
    x5 = c1 + x4
    x6 = c3 / x5
    x7 = x6 - c5
    x8 = x7 * x7
    x9 = x8 * x8
    return x9
```

```
def f10(x):
    c1 = 15141.981176922711
    c2 = -250.55247494972059
    c3 = 5783.5330096027765
    x1 = c3 / x
    x2 = x + x1
    x3 = c1 / x2
    x4 = x3 + x2
    x5 = c2 + x4
    c4 = 501.10494991027866
    x6 = c4 / x5
    c5 = -0.99999999999999956
    x7 = x6 - c5
    x8 = x7 * x7
    x9 = x8 * x8
    x10 = x9 * x9
    return x10
```

*Figure 11.* Discovered programs for exponential ($2^x$) computation, each using a different number of operations. "`def fN(x)`" defines the best program found that uses only N operations.

```
def f7(x):
    c0 = 2.4674011702973546
    c1 = 250.80409460102419
    c2 = 91.207637978888428
    c3 = 2.1131948296213969
    x1 = x * x
    x2 = c2 + x1
    x3 = c1 / x2
    x4 = c3 - x3
    x5 = c0 - x1
    x6 = x5 * x4
    x7 = x4 * x6
    return x7
```

```
def f9(x):
    c0 = -0.49450071689490482
    c1 = -1.1099381478093899
    c2 = -156.23243421346098
    c3 = -0.63212079837291657
    c4 = 32.24336018217511
    c5 = 75.474093940399285
    x1 = x * x
    x2 = c4 - x1
    x3 = c2 / x2
    x4 = c1 + x1
    x5 = c5 / x4
    x6 = x3 - c3
    x7 = x6 - x5
    x8 = x2 / x7
    x9 = x8 - c0
    return x9
```

```
def f11(x):
    c0 = 1.1461229561641841
    c1 = 1565.7032878387506
    c2 = -1047.174821944838
    c3 = 3.4289487947191883
    x1 = x * x
    x2 = c2 - x1
    x3 = c1 / x2
    x4 = c0 + x3
    x5 = x1 * x4
    x6 = x4 * x5
    x7 = c3 - x6
    x8 = x7 * x7
    x9 = x6 * x4
    x10 = x8 * x9
    c4 = 0.9999999999993856
    x11 = x10 + c4
    return x11
```

```
def f13(x):
    c0 = -1635.1048330260724
    c1 = -2.2504924574881073
    c2 = -0.44408342191793626
    c3 = 1.0000000000001437
    c4 = -2.3811015662151589
    x1 = x * x
    x2 = c0 - x1
    x3 = c1 * x1
    x4 = x3 / x2
    x5 = c2 + x4
    x6 = x1 * x5
    x7 = x5 * x6
    x8 = x7 + x4
    x9 = x8 * x5
    x10 = c4 - x9
    x11 = x10 * x10
    x12 = x11 * x9
    x13 = x12 + c3
    return x13
```

```
def f15(x):
    c0 = 0.77843204367750762
    c1 = 9.7631361908027649e-05
    c2 = 1.8362854876663448e-06
    x1 = x * x
    c3 = -0.999999999999994
    c4 = 0.49999999999980405
    x2 = x1 * c4
    x3 = x2 + c3
    x4 = x2 * c2
    x5 = c1 + x4
    x6 = x2 * x5
    c5 = -0.012137371805859434
    x7 = x6 + c5
    x8 = x7 * x3
    x9 = x8 + c0
    c6 = -6.0000000001379634
    x10 = c6 / x2
    x11 = c4 / x9
    x12 = x11 * x11
    x13 = x12 - x10
    x14 = x2 / x13
    x15 = x14 - x3
    return x15
```

```
def f17(x):
    c0 = 0.41257090582954775
    c1 = 0.012760237757718047
    c2 = -5.9999999999999751
    c3 = 6.2152224398902045e-07
    c4 = 1.0084581360976306e-07
    c5 = 0.5
    x1 = c5 * x
    x2 = x * x1
    c6 = -2.0694496116746564e-09
    c7 = -1
    x3 = c2 / x2
    c8 = 0.00018933192802667576
    x4 = x2 + c7
    x5 = x4 * c6
    x6 = c4 - x5
    x7 = x2 * x6
    x8 = c3 + x7
    x9 = x8 * x2
    x10 = x9 - c8
    x11 = x10 * x4
    x12 = c1 - x11
    x13 = x12 * x4
    x14 = x13 - x3
    x15 = x14 + c0
    x16 = x2 / x15
    x17 = x16 - x4
    return x17
```

*Figure 12.* Discovered programs for the cosine computation, each using a different number of operations. "`def fN(x)`" defines the best program found that uses only N operations.

```
                                            def f(x):
                                              c0 = -8.9750363423828947
                                              c1 = -1.6303208576046984
                                              c2 = 0.55281262795996866
                                              c3 = 0.045654921035323662
                                              c4 = -4.895001676862365
                                              c5 = -0.58175574420159881
                                              c6 = 3.5505600959000212
                                              c7 = 2.258822751204606
                                              x1 = x * x
                                              c8 = -3.6339794729813071
        def f(x):                             x2 = x1 - c1
          c1 = 4.0187709297391407             x3 = c6 - x2
          c2 = 1.0627691883670383             x4 = x3 * x
          c3 = 3.5501555239226601             x5 = c7 - x4
          x1 = c3 / x                          x6 = x4 / c5
          x2 = x1 + x                          x7 = x4 - c4
          x3 = c2 / x2                         x8 = x7 * x1
          x4 = x3 + x2                         x9 = x8 - c2
          x5 = c1 / x4                        x10 = x6 * x9
          return x5                          x11 = x10 - c8
                                             x12 = x11 / x2
                                             x13 = x11 * x12
                                             x14 = c0 + x13
                                             x15 = x5 + x14
                                             x16 = x15 / x5
                                             x17 = c3 * x16
                                              c9 = 0.22861498335264485
                                             x18 = x17 + c9
                                             x19 = x15 * x18
                                             return x19
```

*Figure 13.* Discovered code for programs mentioned but not printed in the main text: $f(x) \approx \mathrm{erf}(x)$ for $x \in (0, 2]$ (left) and $f(x) \approx Ai(-7x)$ for $x \in (0, 1]$ (right).

```
def f6(x):
  c1 = -14.790161576419349
  x1 = sqrt(x)
  x2 = x * x
  c2 = -0.50520547016312367
  c3 = 8.4104818471111855
  x3 = c3 + x2
  x4 = c1 / x3
  x5 = c2 - x4
  x6 = x1 / x5
  return x6
```

```
def f7(x):
  c1 = -0.28933343602798889
  c2 = 11.239628874567405
  x1 = sqrt(x)
  x2 = c2 + x1
  c3 = -16.317483851738459
  x3 = c3 / x2
  x4 = c1 - x3
  x5 = x / x4
  x6 = sqrt(x5)
  x7 = x6 / x4
  return x7
```

```
def f8(x):
  c0 = 37.448138217964008
  c1 = -6.5653672504702252
  x1 = x * x
  x2 = c0 + x1
  x3 = sqrt(x)
  c2 = -527.46767212888403
  x4 = c2 / x2
  x5 = c1 - x4
  x6 = x1 / x5
  c3 = 0.79788459329545558
  x7 = c3 + x6
  x8 = x7 * x3
  return x8
```

```
def f9(x):
  c1 = -43.954690020361078
  c2 = -0.16351626858324214
  x1 = sqrt(x)
  x2 = x * x
  c3 = -20.266027650279021
  c4 = 25.219714051793009
  c5 = -9.9026589179219826
  x3 = x2 - c5
  x4 = c4 / x3
  x5 = x3 - c3
  x6 = c1 / x5
  x7 = x4 + x6
  x8 = x7 - c2
  x9 = x1 / x8
  return x9
```

```
def f10(x):
  x1 = x * x
  c1 = 101.66701098222919
  c2 = 0.13298075932556025
  c3 = -4270.3580164057466
  c4 = 28.393794772517104
  x2 = x1 - c1
  c5 = -0.79788456083308135
  x3 = c3 / x1
  x4 = x2 - x3
  x5 = c4 / x4
  x6 = x5 + c2
  x7 = x6 * x1
  x8 = x7 - c5
  x9 = sqrt(x)
  x10 = x8 * x9
  return x10
```

```
def f11(x):
  c1 = -807.78602489533534
  x1 = x * x
  x2 = sqrt(x)
  c2 = -106.06018473018399
  c3 = 0.79788456080275849
  c4 = 4662.5536983828852
  c5 = 14.832695155546126
  x3 = c5 - x1
  x4 = c4 / x3
  x5 = c2 + x4
  c6 = 3.6415712200718193
  x6 = x5 - x1
  x7 = c1 / x6
  x8 = x7 - c6
  x9 = x1 / x8
  x10 = c3 - x9
  x11 = x2 * x10
  return x11
```

*Figure 14.* Discovered code for the modified Bessel function $f(x) \approx I_{1/2}(x)$ for $x \in (0, 1]$. In reading order: 6 operations–11 operations.

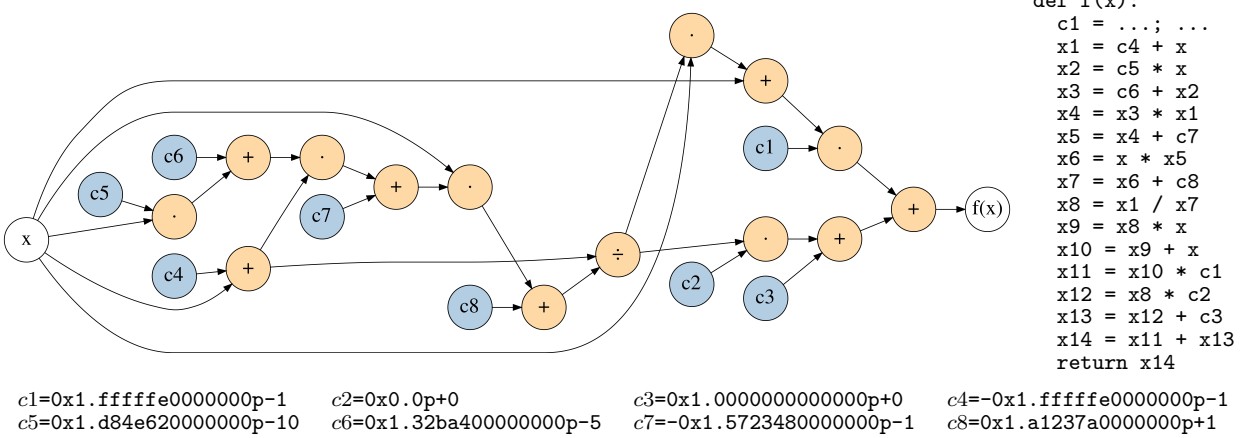

```
def f(x):
  c1 = ...; ...
  x1 = c4 + x
  x2 = c5 * x
  x3 = c6 + x2
  x4 = x3 * x1
  x5 = x4 + c7
  x6 = x * x5
  x7 = x6 + c8
  x8 = x1 / x7
  x9 = x8 * x
  x10 = x9 + x
  x11 = x10 * c1
  x12 = x8 * c2
  x13 = x12 + c3
  x14 = x11 + x13
  return x14
```

$c1$=0x1.fffffe0000000p-1    $c2$=0x0.0p+0    $c3$=0x1.0000000000000p+0    $c4$=-0x1.fffffe0000000p-1
$c5$=0x1.d84e620000000p-10    $c6$=0x1.32ba400000000p-5    $c7$=-0x1.5723480000000p-1    $c8$=0x1.a1237a0000000p+1

*Figure 15.* Discovered program for hardware-aware $2^x$ computation. This program has under 1 ULP of maximum relative `float32` error. With default compiler settings it is more than 3 times faster than the baselines because it triggers a different compilation path.

