# OpenReview forum: "AutoNumerics-Zero: Automated Discovery of State-of-the-Art Mathematical Functions"
_ICML.cc/2026/Conference — ICML 2026 regular_

### Official Review · Reviewer_R4Pf · 2026-03-10

**Soundness:** 3
**Presentation:** 3
**Significance:** 3
**Originality:** 2
**Overall Recommendation:** 4
**Confidence:** 4

**Summary:**

This paper introduces AutoNumerics-Zero, a symbolic regression framework based on evolutionary search, designed to automatically discover compact and accurate computer programs for approximating transcendental functions. The authors test their approach on a range of standard mathematical functions, and show that they achieve higher accuracy than established baselines. The paper provides rigorous mathematical proofs for the error bounds.

**Compliance With Llm Reviewing Policy:**

Affirmed.

**Final Justification:**

The rebuttal clarifies my concerns. I will keep my positive score. The main reason for not raising the score is that, as the authors agree, "the primary contribution—and the main source of our originality—is not algorithmic innovation, but rather an empirical demonstration of the feasibility of a new numerical-design paradigm."

**Key Questions For Authors:**

See the weakness.

**Limitations:**

Yes

**Strengths And Weaknesses:**

Strengths:

1. The empirical results are strong. The discovery of programs that consistently outperform established, highly optimized baselines across multiple functions is a non-trivial and impressive achievement.

2. Nice theoretical results are also given to validate the claims.

3. The method is demonstrated on a diverse set of functions, showcasing its generality.

Weaknesses:

1.  The method is computationally intensive. The authors state they run "100-10k processes for 1-4 days on a custom cluster of Skylake cores"  which represents a significant computational budget. The paper would be strengthened by a more detailed analysis of the computational cost, perhaps in terms of core-hours required to achieve the top results for a single function. How much of the success is due to the algorithmic innovations versus the sheer scale of the search?

2.  Limited novelty in methodological components: The individual components, including genetic programming with graph mutations, NSGA-II for multi-objective optimization, and CMA-ES for coefficient tuning, are all well-established techniques. The main methodological contribution appears to be the careful integration of these components into a distributed, asynchronous framework (dNSGA-II).

3. Dependence on human-crafted range reduction: A critical point, acknowledged by the authors, is that the method relies on the human-provided technique of range reduction (Appendix A.1). The search for an approximation to $2^x$ is not conducted over the entire real line but only over the reduced interval $(0, 1]$. This is a crucial piece of domain knowledge that is injected into the process, meaning the search is not entirely "from scratch." While this is a practical and sensible choice, it does temper the "zero-knowledge" claim. The authors mention this as an avenue for future work (including range reduction in the search), which is appropriate, but it remains a limitation of the current system.

4.  Generalizability of the hardware-aware speedup: The spectacular speedup for the exponential function on Skylake is shown to transfer well to other Intel and AMD architectures (Figure 8). However, it is unclear if this is a general phenomenon or a lucky coincidence. Would a program evolved for speed on one architecture also be fast on a fundamentally different architecture, such as an ARM processor or a GPU?

---

> ### Author Rebuttal · Authors · 2026-03-29
>
> We thank the reviewer for their thoughtful feedback and for recognizing our strong empirical results, the rigorous theoretical validation of our claims, and the generality of our method across diverse functions. We address the reviewer's concerns below.
>
>
> **Compute Cost, Scale, and Amortization**
>
> As suggested by the reviewer, we will add CPU-hour measurements from experiment start to correct program discovery.
>
> Regarding the question of contribution: we agree that scale is indeed a large contributor to our results (as in much of modern machine learning). As we will expand upon in the next section, the novelty of this paper does not center on algorithmic contributions, but rather on the empirical demonstration of a new application paradigm. Here, we draw a direct parallel to the history of Neural Architecture Search (NAS). Early NAS papers relied on highly compute-intensive RL and evolutionary methods to rigorously prove that automated architecture discovery was viable, eventually spurring efficient algorithms like DARTS. We anticipate a similar trajectory here: our paper constitutes the initial step of establishing that evolution can discover state-of-the-art mathematical programs, which we expect will spur future algorithmic improvements to drastically reduce search costs.
>
> To further contextualize the compute budget, we will emphasize two points in the revision. First, the search process represents a strictly one-time upfront cost. Much like the community uses Taylor expansions without re-deriving them, the highly efficient programs discovered by our large-scale search can be freely reused, thus amortizing their search cost (especially in high-performance computing contexts that are bottlenecked by trillions of transcendental function evaluations; e.g., Wu et al. 2023, "Computing and compressing [...]"). Second, we achieved these results using the exact same search method and hyperparameters across our diverse target functions to demonstrate algorithmic robustness. Allowing algorithmic specialization to the real-valued case would likely yield reductions in compute cost.
>
>
> **Methodological Novelty and Originality**
>
> We agree that our individual methodological components—such as genetic programming, NSGA-II, and CMA-ES—are well-established techniques and that our contribution includes their careful integration. However, the originality of our work does not center on algorithmic innovation. Instead, it centers on an empirical systems demonstration of a new numerical-design paradigm: we propose to shift from the traditional focus on arbitrary precision to natively limited-precision approximations, which aligns directly with today's computational reality. By making a state-of-the-art claim in a mathematical domain where machine learning has not traditionally been successfully applied, we demonstrate the plausibility of our proposal. This is in line with the conference's broader view of originality ("originality may arise from [...], application to a real-world use case, or [...]").
>
> Furthermore, we see conceptual novelty in showing that high-quality results can emerge from a data-free search—a powerful, complementary approach to modern Large Language Models (LLMs) that rely on vast amounts of pre-existing human data.
>
> We will update the paper to state this positioning more explicitly.
>
>
> **Dependence on Range Reduction**
>
> We thank the reviewer for highlighting this nuance. We agree that extending our method to the unbounded real line incorporates domain knowledge through range reduction. However, e.g., for the exponential function, any finite combination of basic operations yields a relative error that approaches infinity on the unbounded real line. Range reduction is therefore a mathematical necessity; accordingly, all baselines rely on it too.
>
> In our paper, "zero-knowledge" refers strictly to the automated search process, which looks for an approximation on the bounded reduced interval. The method discovers the core approximation—e.g., a replacement for a truncated Taylor expansion—entirely from scratch. We will update the manuscript to explicitly demarcate the scope of this claim. Furthermore, as noted in Section 4 and agreed upon by the reviewer, incorporating range reduction into the search space is an appropriate direction for future work.
>
>
> **Generalizability of Hardware-Aware Results**
>
> Regarding transferability to fundamentally different architectures (e.g., ARM or GPUs), we share the reviewer's skepticism. The deliberate tradeoff of our hardware-aware approach is that it specializes code to the exact idiosyncrasies of a given architecture family. Therefore, deploying on fundamentally different platforms may naturally require a new search, rather than transferring a Skylake-evolved program directly. Nevertheless, we were very encouraged to find that our speedups did successfully transfer across 8 years of varying Intel and AMD CPU models (Figure 8). We will clarify this in the text.

---

> > ### Author Rebuttal · Reviewer_R4Pf · 2026-04-03
> >
> > As the authors acknowledged, the originality of our work does not center on algorithmic innovation, I keep my positive score.

---

> > > ### Author Response · Authors · 2026-04-06
> > >
> > > We thank the reviewer again for engaging with our rebuttal and for their continued support, particularly their recognition of our "impressive" empirical results and "rigorous" mathematical proofs.
> > >
> > > Regarding the specific point highlighted in their final reply, we completely agree that our methodology relies on what they accurately described as the "careful integration" of well-established components. This characterization aligns with the core point of our rebuttal: our primary contribution—and the main source of our originality—is not algorithmic innovation, but rather an empirical demonstration of the feasibility of a new numerical-design paradigm. To further contextualize this originality, we highlight a central aspect of our framework that we have not yet emphasized in this discussion: the native reuse of intermediate values. As discussed in Section 4 (illustrated in Figure 10 and briefly quantified in Appendix D), unlike traditional, fixed-template baselines (such as Padé or Chebyshev), our program representation natively supports this structural sharing. Because our framework explicitly optimizes for program length alongside accuracy, the evolutionary pressure to minimize the number of operations naturally drives the search to discover programs that reuse intermediate values. This advantage highlights a fundamental distinction from the baseline methods, which, being based on strict mathematical forms, do not naturally support such reuse. Ultimately, this underscores our positioning that natively limited-precision approximations—unconstrained by historical mathematical forms—align directly with today's computational reality, and that machine learning techniques, such as evolutionary search, can plausibly bridge the gap.

---

### Official Review · Reviewer_Fg7S · 2026-03-10

**Soundness:** 2
**Presentation:** 3
**Significance:** 3
**Originality:** 3
**Overall Recommendation:** 5
**Confidence:** 2

**Summary:**

This paper studies automatic approximation of transcendental functions under realistic finite-precision and operation-budget constraints. It proposes AutoNumerics-Zero, a zero-knowledge symbolic-regression framework that starts from empty programs and uses evolutionary search plus coefficient optimization to discover compact arithmetic code for function approximation from scratch, rather than relying on fixed templates such as Taylor or Padé forms.

Empirically, the paper shows that the discovered programs can outperform strong classical baselines for several target functions in the high-but-finite accuracy regime, including notable results for exponential approximation with certified error bounds, and also demonstrates a hardware-specialized variant that trades generality for speed. Overall, the contribution is a practical demonstration that large-scale symbolic program search can uncover novel and useful numerical approximations beyond standard hand-designed families.

**Compliance With Llm Reviewing Policy:**

Affirmed.

**Final Justification:**

The rebuttal addressed most of my main concerns and improved my overall assessment of the paper.  Given the originality of the idea, the practical significance of the problem, and the improved confidence provided by the rebuttal, I have decided to raise my final score.

**Key Questions For Authors:**

1. How general are the conclusions beyond the six target functions studied here?

The current results are strong case studies, but it is unclear how broadly the advantage extends. Additional results on a more systematic benchmark would strengthen the paper’s significance.

2. Can the authors better quantify the search cost versus deployment benefit?

The discovered approximations are impressive, but the search cost is very high. A clearer amortization analysis, or sensitivity to compute budget, would help assess practical impact.

3. How much tuning is required per target function?

It would be helpful to clarify how much the final gains depend on target-specific choices such as operation sets, search space design, and baseline selection. This would affect how broadly useful the method is in practice.

4. How portable are the hardware-aware results?

The hardware-specialized experiment is interesting, but it is unclear how much of the gain transfers across architectures or compiler stacks. This would help calibrate the broader applicability of the strongest systems result.

5. How should the contribution be positioned with respect to originality?

 The paper is clear that it does not introduce a new search algorithm, which I appreciate. A more explicit statement of whether the main contribution is a new numerical-design paradigm or a strong empirical systems demonstration would help calibrate the novelty claim.

**Limitations:**

Partially. The paper is reasonably transparent about some important limitations, especially that it does not claim universal improvement across all functions and that the hardware-aware results trade generality for efficiency.
However, I think the limitations discussion should more explicitly emphasize the very high search cost and the resulting reproducibility/accessibility barrier, since this materially affects the practical value of the approach.

**Strengths And Weaknesses:**

**Soundness**

The paper is technically solid for the claims it makes. The method is clearly specified as a combination of program search, NSGA-II-style selection, graph mutation, and CMA-ES coefficient tuning, and the main empirical claims are supported by strong case studies, mathematical error certification for the exponential approximation, and component ablations.    The main weakness is not correctness but scope: the evaluation is based on six target functions, and the paper explicitly does not claim broad dominance across all functions.

**Presentation**

The paper is clear and well written. The motivation is easy to follow, the method is described concretely, and the results are presented in a way that makes the practical contribution clear. I also appreciate that the authors are explicit that this is not a new search algorithm, but rather a system-level demonstration built from known components.  A minor weakness is that the discussion sometimes reads more strongly than the actual experimental breadth justifies.

**Significance**

I think the problem is meaningful. Searching directly for finite-precision approximations under realistic operation or hardware constraints is a useful perspective, and the exponential and hardware-aware results are genuinely interesting. The paper shows that symbolic program search can produce approximations that outperform strong classical baselines in some practically relevant regimes.   At the same time, the practical impact is moderated by the very high search cost: the authors report running 100–10k processes for 1–4 days and estimate reproduction would require roughly 100 modern high-end CPUs.

**Originality**

The paper is moderately original. The main novelty is not algorithmic, but in applying large-scale zero-knowledge symbolic program search to transcendental-function approximation and showing that unconventional discovered programs can beat standard hand-designed families at finite precision.   This is interesting, but I would still view the contribution as a strong empirical/system paper rather than a major methodological advance.

---

> ### Author Rebuttal · Authors · 2026-03-29
>
> We thank the reviewer for their thoughtful feedback and for recognizing our paper as a strong empirical systems demonstration that tackles a meaningful problem. We address the reviewer's concerns below.
>
>
> **Generality Beyond the Six Target Functions**
>
> As the reviewer notes, we do not claim broad dominance. We emphasize, however, that these targets were chosen *a priori* to represent a diverse set of numerical challenges (e.g., Airy's oscillations, Bessel's singularity, erf's slow convergence). In particular, we did not exclude search experiments because they failed; the closest the method came to struggling was transparently reported (the error function). We will consolidate this selection rationale into the Results introduction.
>
>
> **Search Cost and Deployment Benefit**
>
> Regarding amortization, we consider deployment in high-performance computing (e.g., quantum chemistry or climate modeling), where transcendental functions are often the bottleneck and are evaluated trillions of times (e.g., Wu et al. 2023, "Computing and compressing [...]"). In such domains, the one-time search cost is rapidly amortized, as even moderate execution speedups yield substantial computational savings at scale.
>
> We agree the compute-intensive search poses an accessibility barrier. However, it is important to distinguish between accessibility and reproducibility: while the requisite resources are not universally accessible, independent repeats of our search with the same resources consistently yield comparable results (Appendix D), demonstrating algorithmic robustness. Furthermore, regarding accessibility, the discovered programs themselves are readily accessible; they can be shared without redoing the search.
>
> Finally, we view this high-compute demonstration as a starting point, drawing a direct parallel to the history of Neural Architecture Search (NAS). High-compute RL and evolutionary NAS experiments served as initial milestones proving viability, eventually spurring efficient algorithms like DARTS. We anticipate a similar trajectory here, with our paper establishing the initial viability.
>
> We will add this context to a new Limitations section.
>
>
> **Tuning per Target Function**
>
> AutoNumerics-Zero requires minimal tuning. Search hyperparameters (e.g., population size, mutation rates) and search space design were kept strictly constant across all functions. The only target-specific choice was the addition of the square root to the operation set of the Bessel function, a deliberate a-priori choice to illustrate the framework's extensibility. Regarding baseline selection, we tailored our comparisons to provide strong competition for each function. For our main result (the real-valued exponential), we exhaustively evaluated every established baseline we could identify. For the remaining real-valued targets, we focused on those expected to perform best (rationale documented in each section). Finally, for the hardware-aware case, we evaluated a broader set, incorporating common practices, strong performers from earlier sections, and the strongest modern floating-point-specialized method we could identify. Importantly, we present every baseline evaluated. We will incorporate these points.
>
>
> **Portability of Hardware-Aware Results**
>
> Regarding hardware portability, as detailed in Section 3.6 and Figure 8, the speedups discovered on an Intel Skylake architecture successfully transfer across 8 years of varying Intel and AMD CPU models, retaining an 80% to 300% speedup over the top baselines. To ensure this finding is not missed, we will reference Figure 8 more prominently in the text. We acknowledge that we did not explore multiple compiler stacks due to infrastructure constraints, which is a limitation we will note.
>
>
> **Originality and Positioning**
>
> To clarify our positioning: we view our core conceptual motivation as proposing a new numerical-design paradigm (shifting from arbitrary precision to natively limited-precision approximations, to align with today's computational reality). However, we agree that our immediate contribution is best categorized as a strong empirical systems demonstration. As the reviewer notes, our paper explicitly states that we do not introduce a new search algorithm, nor do we claim broad dominance across all functions. Rather, by achieving state-of-the-art results on these six diverse targets, we empirically demonstrate the *plausibility* of this new limited-precision approach. This aligns with the conference's broader view of originality ("originality may arise from [...] application to a real-world use case, or [...]"), as we set a precedent in a mathematical domain where machine learning has not traditionally been successfully applied. We will update the Discussion to explicitly reflect this distinction, ensuring our tone is carefully calibrated to our experimental breadth, so that our work can serve as a grounded starting point for the community.

---

> > ### Author Rebuttal · Reviewer_Fg7S · 2026-04-03
> >
> > Thank you for the thoughtful and technically detailed rebuttal; it clarifies several important points and improves my understanding of the method, although it does not fully resolve my main concerns.

---

> > > ### Author Response · Authors · 2026-04-06
> > >
> > > We thank the reviewer for their constructive engagement and for confirming that our initial rebuttal clarified technical details of our method. To further address their original question regarding the paper's positioning, we wish to highlight a central aspect of our framework that we omitted from our initial rebuttal: the native reuse of intermediate calculations. Unlike traditional baselines (such as Padé or Chebyshev expansions) that rely on fixed mathematical templates, our approach naturally engineers intermediate value reuse into the discovered programs, which minimizes operation counts (illustrated in Figure 10; briefly quantified in Appendix D). This is possible because our zero-knowledge framework starts from empty programs and operates from scratch—without using expansion rules or pre-training on human-designed functions. Left free from the bias of existing approaches, the evolutionary pressure to reduce the number of operations naturally drives the search to discover and exploit these structural efficiencies. This structural advantage is key to the originality of our proposed numerical-design paradigm, as it is a direct contributor to the efficiency of the discovered programs. We will highlight this in the revised Discussion section and tie it directly to these efficiency gains. Ultimately, this situates our work as an empirical demonstration of a practical capability lacking in previous methods: the ability to natively maximize computational efficiency through intermediate calculation reuse.
> > >
> > > Finally, we fully acknowledge the reviewer's remaining concerns regarding the high search costs and the evaluation scope. While our initial rebuttal details the mitigating context and rationale for these practical boundaries, we agree that their explicit documentation is essential. Therefore, we will codify them in the Results introduction and a new Limitations subsection of the Discussion section.

---

### Official Review · Reviewer_Xkyu · 2026-03-10

**Soundness:** 3
**Presentation:** 3
**Significance:** 3
**Originality:** 3
**Overall Recommendation:** 5
**Confidence:** 4

**Summary:**

This is a paper that uses symbolic computation to find finite-precision approximation of certain continuous functions where error is measured only by maximum relative error. The paper also includes hardware-aware variants of the main idea.

**Compliance With Llm Reviewing Policy:**

Affirmed.

**Final Justification:**

I changed my rating reflecting the discussion and I think this rating is fair.

**Key Questions For Authors:**

Does your paper involve any machine learning?  Or is it a straight-forward application of an evalutionary algorithm?

Why is maximal relative error alone a good error measure for your approximation without incorporating other standard considerations (computaitonal cost, stability) in numerical analysis?

What are the limitations of your method?

**Limitations:**

Understanding limitations would require more experimentation that is not present in the paper, so I don't have any good insight on the limitations of the proposed method.

**Strengths And Weaknesses:**

Strengths:  The paper adresses an important problem. It takes advantage of current advancements in evalutionary algorithms, particularly the Symbolic regression package, in this sense it is timely. The appendix is relatively detailed and clear.

Weaknesses:  1) First weakness in my view is a technical one: Measuring the quality of an approximation only with relative error is too simplistic. One needs to look at two more aspects (at least):  the cost of approximation, and the stability of approximation. The later is usually analyzed using condition numbers, and is subtle. (see e.g. https://arxiv.org/abs/2109.10610 for some points on subtleties of stability analysis).     2) Second weakness is the mismatch of venue: the paper doesn't really have learning in it, and ICML is a machine learning conference. My impression is that this paper would be a better fit to venues on scientific computing/ numerical analysis.

---

> ### Author Rebuttal · Authors · 2026-03-29
>
> We thank the reviewer for the constructive feedback, and for recognizing the timeliness and importance of the problem our paper tackles. We address the reviewer's concerns below.
>
>
> **Relevance to ICML and Machine Learning**
>
> We thank the reviewer for raising the important question of venue fit. Our work aligns directly with the ICML 2026 Call for Papers, which explicitly welcomes "application-driven machine learning" and methodologies beyond deep learning, specifically including stochastic optimization. Symbolic regression and evolutionary search algorithms---the central methods of our paper---are foundational stochastic optimization methodologies within machine learning, particularly in the subfields of representation learning, black-box optimization, and automated algorithm discovery. Applying these core techniques to discover novel mathematical programs is a relevant application-driven contribution. We will revise our introduction to more explicitly frame our methodology within the broader machine learning context.
>
>
> **Evaluation Metrics: Cost, Stability, and Maximum Relative Error**
>
> We agree with the reviewer on the critical importance of considering computational cost and stability. We wish to clarify that computational cost is already a primary metric in our experiments. Indeed, our dual-objective evolutionary search explicitly optimizes for cost alongside accuracy to produce the Pareto fronts presented in our results. Specifically, for our real-valued results (Sections 3.1–3.5), we use the number of operations as the cost metric. We chose this metric to align with established baselines, many of which predate widespread use of computers; e.g., Macon's continued fractions, in fact, explicitly minimize operation counts. Furthermore, for our float-valued hardware-aware results (Section 3.6), we instead used the wall-clock execution time required by a CPU core. This highly practical choice intentionally highlights a key advantage of our method: because AutoNumerics-Zero is a black-box optimizer, it can directly handle complex, non-differentiable cost objectives---even the inner workings of a CPU---that are virtually impossible to optimize using traditional mathematical approaches or gradient descent.
>
> The numerical stability of our float-valued programs is rigorously guaranteed by our mathematical error proofs. As detailed in Appendix B.3, we use the GAPPA automated prover to establish strict upper bounds on the error of our hardware-aware programs while rigorously accounting for IEEE-754 floating-point rounding at every intermediate operation. If a discovered program were numerically unstable, the accumulated floating-point errors would explode, and the interval arithmetic proof would simply fail to bound the error under 1 ULP. Because our proofs succeed in bounding the output of the fully executed program across the domain, stability is mathematically encompassed. We will add a specific note in Section 3.6 clarifying how these interval arithmetic proofs guarantee stability.
>
> Regarding our accuracy metric, we prioritize maximum relative error (alongside the aforementioned cost and stability guarantees) because it is the gold standard for this problem in the numerical analysis literature (e.g., Hart 1978, "Computer approximations."). Using the maximum relative error ensures that our accuracy claims are rigorous and directly comparable against established baselines.
>
>
> **Limitations of the Proposed Method**
>
> We agree that our manuscript would benefit from a more explicit discussion of the method's limitations. To address this, we will expand the *Discussion* with a dedicated *Limitations* subsection covering the following.
>
> First, our method relies on range reduction. However, all baselines rely on it too. This is because, for example in the exponential function's case, any finite combination of basic operations results in unbounded error. Nevertheless, Section 4 notes that future work may incorporate range reduction directly into the search space.
>
> Second, the search process requires substantial upfront computational resources. We will clarify that this is a one-time cost amortizable by the repeated use of the discovered programs, which in high-performance computing contexts may be run trillions of times (e.g., Wu et al. 2023, "Computing and compressing [...]"). Furthermore, we view this compute-intensive demonstration as a starting point, drawing a direct parallel to the history of neural architecture search (NAS). NAS was pioneered using highly compute-intensive RL and evolutionary methods; these early search experiments were crucial milestones that proved viability and ultimately inspired highly efficient algorithms like DARTS. We anticipate a similar trajectory here: establishing that symbolic regression discovers state-of-the-art mathematical programs is an initial step, which we expect will spur future algorithmic improvements to reduce the upfront search cost.

---

> > ### Author Rebuttal · Reviewer_Xkyu · 2026-03-31
> >
> > The authors explanation on cost being included improved my understanding of their work.   The stability aspects remain unaddressed. I appreciated their acknowledgement of fair consructive suggestions, e.g., the lack of explanation of limitations.

---

> > > ### Author Response · Authors · 2026-04-06
> > >
> > > We thank the reviewer for their continued engagement and are glad that our explanation of the cost metrics was helpful. We also appreciate the reviewer's concern on the subtlety of stability, as it is indeed a critical aspect of numerical analysis. We realize that our initial manuscript did not sufficiently highlight how our method guarantees stability, so we take this opportunity to provide more detail. We will update the main text to make the following explicit.
> > >
> > >
> > > **Forward Stability and Intermediate Errors**
> > >
> > > In traditional numerical analysis, as the reviewer pointed out in their initial review, stability is often analyzed using condition numbers. However, because our method produces explicit, finite computer programs (on the reduced interval), we are able to take a different, highly rigorous approach that avoids the need for condition-number estimations: we mathematically prove a *uniform* bound on the end-to-end relative error while accounting for the maximum possible floating-point rounding error at each intermediate step. (Beyond the reduced interval, stable range reduction methods exist independently of our paper, e.g., Defour et al. 2001, "A new range reduction algorithm").
> > >
> > > As detailed in Appendix B.3, because our proofs bound the final output of the float-valued fully executed program under 1 ULP across the domain, forward stability is mathematically guaranteed. To establish this, we utilize the Gappa automated prover, which was specifically designed to perform forward error analysis over floating-point arithmetic. As the Gappa paper states, the tool "uses multiple-precision dyadic fractions for the endpoints of intervals and performs forward error analysis on rounded operators" (Daumas and Melquiond, 2010, "Certification of bounds on expressions involving rounded operators.").
> > >
> > > Crucially, this end-to-end bound guarantees both the stability of the approximation under finite-precision arithmetic and its mathematical correctness. Regarding the finite-precision stability, a critical subtlety of forward error analysis—and a common source of instability—is the potential for errors to accumulate at intermediate steps due to the loss of precision from floating-point rounding. The Gappa prover directly accounts for this, as it simulates IEEE-754 rounding at every intermediate calculation, compounding the worst-case rounding error at every individual operation in the program's compute graph, and propagating these bounds forward to the output node. Furthermore, regarding mathematical correctness, a program could theoretically perform error-free intermediate calculations yet arrive at the wrong answer due to a poor mathematical approximation; our proof precludes this by strictly bounding the total error against the true target function. The use of this tool avoids analyzing the formulas purely algebraically, which would quickly become cumbersome.
> > >
> > > Finally, while the main subtlety of stability lies in the finite-precision arithmetic described above, we also mathematically prove these uniform error bounds for our real-valued expressions (Appendix B.2). Because real-valued expressions do not suffer from intermediate floating-point rounding errors, their forward stability is a more direct consequence of the established relative error bounds, and can be justified with a similar but simpler argument to the one above.
> > >
> > >
> > > **Future Work: Backward Stability**
> > >
> > > Because our evolutionary process explicitly optimizes for accuracy, proving a uniform bound on the relative error (forward stability) was the most natural choice. However, as the reviewer alluded to with their reference to condition numbers, exploring backward stability is another valid and rigorous lens in numerical analysis. We hypothesize that our evolutionary method could be adjusted to optimize for functions that are backward-stable; we leave this exploration to future work.
> > >
> > > Motivated by this discussion, we will explicitly detail the mechanics of our Gappa proofs in the main text, and we will add a dedicated paragraph to our Discussion section addressing our choice of forward stability and the potential for exploring backward stability in future work.

---

### Official Review · Reviewer_6Knu · 2026-03-11

**Soundness:** 3
**Presentation:** 3
**Significance:** 3
**Originality:** 3
**Overall Recommendation:** 4
**Confidence:** 3

**Summary:**

AutoNumerics‑Zero is a zero‑knowledge symbolic‑regression system designed to automatically discover compact, high‑accuracy approximations of transcendental mathematical functions such as the exponential, cosine, Airy, Bessel, and error functions.The system evolves mathematical programs from scratch using only basic operations {+, −, *, /} and without injecting any mathematical priors, templates, or symbolic transformations. Through a nested evolutionary process combining distributed NSGA‑II for program structure and CMA‑ES for coefficient tuning, the method automatically generates novel computer programs.

The system discovered a 10‑operation approximation of 2^x that guarantees 14 significant figures of accuracy, exceeding the precision of previously known formulas of similar size by over six orders of magnitude.The method can be specialized to particular hardware architectures (e.g., Intel Skylake), yielding implementations that achieve over 3× speedups compared to baselines.

**Compliance With Llm Reviewing Policy:**

Affirmed.

**Key Questions For Authors:**

1. Given the extremely high computational cost, how do the authors envision reproducibility of the method?
2. How do the authors propose ensuring the interpretability, verifiability, and long‑term maintainability of the discovered expressions?

**Limitations:**

yes

**Strengths And Weaknesses:**

Strengths:
1. The presentation style is clear, structured, and supported with illustrative figures namely mutation graphs, Pareto‑front plots, and discovered programs which guide the readers.
2. The significance of the work lies in the zero‑knowledge symbolic regression method that outperforms approximation techniques, producing expressions that achieve higher accuracy with fewer operations.

Weaknesses:
1. The method is resource‑intensive and requires access to high end specialised hardwares making it difficult to reproduce.
2. The method lacks interpretability and transparency as the evolved structures would be difficult to reformulate or validate manually.

---

> ### Author Rebuttal · Authors · 2026-03-29
>
> We thank the reviewer for the constructive feedback, and for recognizing the clarity of our presentation and the significance of our zero-knowledge symbolic regression method. We address the reviewer's concerns below.
>
> **Reproducibility and Computational Cost**
>
> The reviewer notes that our resource usage can pose a barrier to replication of the results. To address this, we distinguish between reproducing the discovered programs and reproducing the search process itself:
>
> * Regarding the discovered programs, our claims are easily and immediately reproducible. We have open-sourced the programs, and the automated proofs in our appendix are explained in full. We will add to the paper the exact scripts to reproduce our use of provers.
>
> * Regarding the search process, we agree there is value in independent teams re-running the experiments, and that compute can pose an accessibility barrier. However, reproducibility—as opposed to accessibility—fundamentally requires that the same results be reliably obtained given the same resources, and holds even if those resources are not *universally* accessible (as often happens with large models today). To demonstrate this robustness, we conducted independent repeats of our search, which consistently yield comparable results (Appendix D). Finally, regarding accessibility, we anticipate future efficiency improvements analogous to the history of neural architecture search (NAS). Early compute-intensive RL and evolutionary NAS experiments served as milestones proving viability, while follow-ups like DARTS vastly improved efficiency. We expect a similar trajectory here, with this paper demonstrating initial viability by discovering state-of-the-art programs.
>
>
> **Verifiability of Discovered Programs**
>
> The reviewer asks how we ensure the verifiability of the discovered expressions. We appreciate the reviewer raising this crucial point; it highlights that we have not sufficiently emphasized our verification guarantees in the main text. Our discovered programs are fully verifiable, as demonstrated for our most salient results through our mathematical proofs in Appendix B. The appendix rigorously verifies multiple discovered programs through highly mechanical proofs that can be re-applied to future results. For our real-valued programs, we do this by using automated interval arithmetic through the IBEX library to establish rigorous Lipschitz bounds on the derivatives of the discovered functions, and rely on their differentiability to establish strict upper bounds on the error. For our hardware-aware programs, we use the GAPPA prover to rigorously bound accumulated IEEE-754 floating-point rounding errors. We will elevate a summary of these proofs to the main text so readers immediately understand that our most surprising discovered algorithms are fully verified.
>
>
> **Interpretability**
>
> We agree that the evolved structures are difficult to interpret. Indeed, there is a tradeoff between interpretability and the algorithmic quality of the results. Traditional approximation methods (e.g., Taylor or Padé) are highly interpretable because they conform to a rigid, hand-designed structure (e.g., polynomial or rational). AutoNumerics-Zero intentionally discards these mathematical priors. By searching over the unconstrained space of basic operations, the system discovers novel, unconventional forms that aggressively reuse intermediate values. This structural flexibility is precisely why our 10-operation approximation of 2^x can guarantee 14 significant figures of accuracy, exceeding previously known formulas of similar size by over six orders of magnitude. This paradigm shift is somewhat analogous to the machine learning transition from hand-designed features to deep learning: neural network weights are notoriously more difficult to interpret than hand-designed feature extraction rules, but the approach compensates for its obscurity through state-of-the-art performance. We will add a discussion of this tradeoff to Section 4.
>
> **Long-term maintainability**
>
> The reviewer asks how we ensure the long-term maintainability of these expressions. This is an important point but we wish to clarify that the burden of hardware-specific maintenance applies to only a small fraction of our paper. The vast majority of our results (Sections 3.1 through 3.5) evaluate real-valued, hardware-independent functions that require no maintenance. The hardware-specific superoptimization the reviewer refers to is confined to Section 3.6, which targets the float32 type on Intel Skylake processors. The accuracy of these specific floating-point values will remain strictly true for as long as the IEEE float32 standard is used. While execution speed is inherently hardware-dependent, we specifically demonstrated in Figure 8 that our discovered programs maintain significant speedups across 8 years of different AMD and Intel CPU architectures.

---

> > ### Author Rebuttal · Reviewer_6Knu · 2026-04-01
> >
> > Thank you for the detailed explanations regarding the reproducibility and verifiability of the discovered programs. Given the short turnaround time, it is understandable that identifying hardware-agnostic mathematical expressions could not be fully addressed. However, incorporating a discussion on interpretability in the context of discovering mathematical expressions would significantly strengthen the paper. As these aspects remain unresolved, I will maintain my original score.

---

> > > ### Author Response · Authors · 2026-04-06
> > >
> > > We thank the reviewer for their continued engagement and for the helpful suggestions. Below we address the two remaining points.
> > >
> > >
> > > **Identifying hardware-agnostic mathematical expressions**
> > >
> > > We apologize for any lack of clarity on our end regarding the hardware-agnostic nature of the results in the various sections. We are revising the paper to make the distinction between the hardware-agnostic and hardware-specific results explicitly clear:
> > >
> > > * We are adding text to Sections 3.1–3.5 to emphasize that all mathematical expressions discovered in these sections are inherently hardware-agnostic. That is, they are hardware-agnostic in the exact same sense that the baselines (e.g., Taylor expansions) are hardware-agnostic.
> > >
> > > * We are also adding a preamble to Section 3.6 ("hardware-aware [...]") highlighting that this is the only section where we intentionally trade hardware-agnostic generality to optimize efficiency for a specific hardware family. Thus, this section is—*by design*—not hardware-agnostic (though Figure 8 demonstrates transferability within the family).
> > >
> > >
> > > **Incorporating a discussion on interpretability**
> > >
> > > As suggested by the reviewer, we are adding a dedicated subsection on interpretability in the paper's Discussion (Section 4). This new text will detail the trade-off between the highly interpretable, rigid structures of traditional methods (e.g., Taylor expansions, Padé approximants) and the less interpretable, unconstrained forms discovered by AutoNumerics-Zero. We will note the benefits of interpretable forms (e.g., Taylor expansions provide an appealing structure as a sum of terms representing increasingly finer corrections). We will also emphasize that in some contexts, the loss of structural interpretability can be justified by significant efficiency and accuracy gains. For instance, sacrificing structural priors allowed our system to discover a 10-operation approximation of $2^x$ that guarantees 14 significant figures of accuracy, exceeding previously known formulas of similar size by over six orders of magnitude. We will illustrate the trade-off by drawing an analogy to the history of machine learning, where the field transitioned from interpretable hand-designed features to highly performant, yet opaque, deep learning weights. This framing properly highlights the respective merits of both the traditional baselines and our approach.

---

### Decision · Program_Chairs · 2026-04-30

**Decision:**

Accept (regular)

**Comment:**

The paper proposes an evolutionary strategy to search for 'optimal' programs for basic functions. Evolutionary methods for symbolic regression have shown great potential. For example, a 10-step program that approximate the exponential function with 14 digits. All reviewers are positiveю